# DAGFormer: A graph-based domain adaptation approach for single-cell cancer drug response prediction

Fen Yan[1], ZhiHua Du[1]*, Yu-An Huang[2,3]*

**1** College of Computer Science and Software Engineering, Shenzhen University, Shenzhen, China,
**2** School of Computer Science, Northwestern Polytechnical University, Xi'an, China, **3** Research &
Development Institute of Northwestern Polytechnical University in Shenzhen, Shenzhen, China

* duzh@szu.edu.cn (ZD); yuanhuang@nwpu.edu.cn (YH)

DAGFormer: A graph-based domain
adaptation approach for single-cell cancer
drug response prediction. PLoS Comput
Biol 21(12): e1013832. https://doi.org/10.
1371/journal.pcbi.1013832

School of Medicine, UNITED STATES OF
AMERICA

**Peer Review History:** PLOS recognizes
the benefits of transparency in the peer
review process; therefore, we enable the
publication of all of the content of peer
review and author responses alongside
final, published articles. The editorial
history of this article is available here:
https://doi.org/10.1371/journal.pcbi.
1013832

## Abstract

Developing computational methods for single-cell drug response prediction deepens
our understanding of tumor heterogeneity and uncovers resistance mechanisms critical to improving cancer therapy. However, current approaches struggle to fully capture intratumoral heterogeneity, as bulk RNA sequencing (bulk RNA-seq) obscures
heterogeneity across individual cells, while single-cell RNA sequencing (scRNA-seq)
remains constrained by limited throughput and high cost. Current approaches integrating bulk and scRNA-seq data frequently encounter batch effects, impairing robust
knowledge transfer. Moreover, most existing methods overlook the role of intercellular interactions, treating cells as isolated entities. To overcome these limitations, we
propose DAGFormer, a Graph-based Domain Adaptation framework that integrates
bulk and scRNA-seq data for predicting single-cell drug responses. DAGFormer constructs cellular neighbor graphs using diverse topological strategies and employs
Graph Domain Adaptation (GDA) to bridge graph-level distribution gaps between bulk
and single-cell RNA-seq data. A dual-domain decoder further disentangles shared
and modality-specific representations, preserving both general and unique biological signals. Benchmarking DAGFormer on ten independent scRNA-seq datasets
demonstrated its superior performance compared to existing methods, underscoring
its effectiveness and robustness in cancer drug response prediction.

## Author summary

In the era of precision medicine, single-cell RNA sequencing (scRNA-seq)
provides gene expression profiles at the individual cell level, while bulk RNA
sequencing (bulk RNA-seq) captures the averaged transcriptome of mixed cell
populations. Predicting drug sensitivities at the single-cell level can offer valuable insights into the mechanisms of treatment response heterogeneity and drug
resistance. Few

**Data availability statement:** Python code and the datasets used in our studies are made available at

https://github.com/yanfen-git/DAGFormer.

**Funding:** This work was supported by the National Key R&D Program of China (Grant 2020YFA0908700 to Z.D.); the National Natural Science Foundation of China (Grants 62176164 to Z.D., 62522316 to Y.H., and 62472353 to Y.H.); the Natural Science Foundation of Guangdong Province (Grant 2023A1515010992 to Z.D.); the Guangdong Basic and Applied Basic Research Foundation (Grant 2024A1515011984 to Y.H.); and the Science and Technology Innovation Committee Foundation of Shenzhen (Grant JCYJ20220531101217039 to Z.D.). The funders had no role in study design, data collection and analysis, decision to publish, or preparation of the manuscript.

**Competing interests:** The authors have declared that no competing interests exist.

studies to date have successfully integrated bulk RNA-seq and scRNA-seq data for drug response prediction, and only a limited number of computational methods have achieved promising results. However, this computational approach has a key limitation: drug response at the single-cell level is not solely determined by the intrinsic gene expression of individual cells, but is also influenced by intercellular interactions within the tumor microenvironment. To overcome the above limitation, we propose DAGFormer, a novel computational framework. The core idea of DAGFormer is to construct cellular neighbor graphs using different topological strategies and apply Graph Domain Adaptation (GDA) to reduce the distributional gap between cell relationship graphs derived from bulk and single-cell RNA-seq data. The experimental results show that by constructing cellular relationship graphs, DAGFormer effectively addresses the inherent batch effects between cell graphs derived from bulk and single-cell RNA-seq data, and accurately predicts cancer drug responses at the single-cell level.

## 1 Introduction

In the era of precision medicine, tailoring cancer treatment to the genomic profiles of individual cells holds great promise [1]. Although advances in genomic profiling and in vitro drug screening have generated extensive pharmacological response data across various cancer cell lines, clinical outcomes remain suboptimal due to frequent cancer relapse, which is primarily driven by intratumoral heterogeneity (ITH) [8–10]. ITH primarily manifests in two dimensions: genetic heterogeneity [11,21,22], where distinct subclones within the same tumor harbor different combinations of mutations; and cellular diversity [20], characterized by the coexistence of multiple subpopulations with divergent origins and functional traits. This complexity allows inherently resistant subpopulations or adaptive cells to survive treatment, seeding minimal residual disease that ultimately triggers recurrence [26].

During therapy, anticancer agents often eliminate only drug-sensitive clones, while pre-existing resistant subpopulations evade eradication and subsequently expand [23,24]. Such intrinsic resistance typically arises from pre-treatment genetic alterations. For instance, in high-grade serous ovarian cancer, TP53 mutations occur in approximately 96% of cases, often accompanied by deficiencies in DNA repair genes such as BRCA1/2 and RAD51C, conferring survival and drug resistance advantages [27]. Similar mechanisms have been documented in malignancies such as lung cancer [15,16], breast cancer [13], and glioblastoma [12,14]. Additionally, drug treatment itself may also induce the formation of new drug resistance in otherwise sensitive tumour cells [19]. For example, prolonged EGFR inhibitor treatment in lung cancer can activate AXL-mediated signaling, inducing epithelial-to-mesenchymal transition and promoting resistance [18]. In melanoma, BRAF inhibitor therapy can trigger secondary NRAS or MEK mutations, leading to compensatory MAPK pathway

activation and tumor regrowth [17]. To elucidate drug resistance mechanisms more precisely, we propose stratifying drug response data into two temporal categories: pre-treatment and post-treatment. Pre-treatment data help uncover innate resistance pathways and inform initial therapeutic design, whereas post-treatment data capture dynamic tumor evolution under therapeutic pressure, supporting timely adjustment of treatment regimens.

Current cancer drug response prediction models primarily rely on bulk RNA sequencing (bulk RNA-seq) data, which aggregate gene expression from all cell types within a tumor. However, this approach masks the heterogeneity among individual cells, limiting the ability of bulk-based analyses to accurately predict the response of specific tumor cell subpopulations to treatment and to identify rare drug-resistant clones [28]. The limitations of bulk RNA-seq have been experimentally validated in studies such as those by Kim et al., who demonstrated that transcriptional signatures of drug-resistant subpopulations in lung adenocarcinoma are diluted in bulk RNA-seq data but become clearly identifiable in single-cell RNA sequencing (scRNA-seq) [25]. This finding underscores the challenge of detecting rare resistant clones using bulk-based analyses. In contrast, scRNA-seq provides a finer-resolution perspective on the heterogeneous drug responses of tumor subpopulations [30]. However, the application of scRNA-seq in drug response prediction remains constrained by high costs and limited sample sizes [31]. Integrating bulk RNA-seq, which provides a comprehensive tumor-wide perspective, with scRNA-seq, which offers single-cell resolution, represents a promising strategy for enhancing drug response prediction at the single-cell level [32].

In recent years, deep learning (DL) methods have made significant progress in integrating bulk RNA-seq and scRNA-seq data [29]. DL methods primarily utilize neural networks to integrate bulk RNA-seq and scRNA-seq data. These methods minimize distributional differences between the two datasets through techniques such as transfer learning. For example, scDEAL employs deep transfer learning to integrate bulk and single-cell RNA-seq data for cancer drug response prediction [33]. It uses a denoising autoencoder for feature selection and applies maximum mean discrepancy (MMD) minimization to reduce distributional shifts across data sources, ensuring effective data fusion. SCAD leverages adversarial learning, using a domain discriminator to mitigate cross-source biases and extract invariant features, thereby enabling data integration [34].

Recently, more specialized DL frameworks have emerged to tackle increasingly complex challenges in this domain, with a focus on either enhancing biological feature quality or refining domain adaptation techniques. For instance, models such as scAdaDrug [4] emphasize advanced multi-source domain adaptation to improve transfer robustness, while scGSDR [5] enhances predictive power by explicitly incorporating gene semantics and pathway knowledge to produce more informative feature representations. These recent developments illustrate a shift toward biologically informed and structurally adaptive deep learning strategies, setting the stage for further innovations such as DAGFormer.

Despite the advancements in cancer drug response prediction, several unresolved challenges remain. First, drug response at the single-cell level is not solely determined by the intrinsic gene expression profiles of individual cells. Instead, it is significantly regulated by intercellular interactions within the tumor microenvironment [3,6,7]. However, most existing studies treat cells as independent entities and fail to account for the impact of cell-cell interactions on drug sensitivity. Constructing a topological representation of cellular relationships, such as a graph structure, may enhance the reliability of drug response prediction.Second, due to differences in sequencing platforms, dataset scales, and data processing methodologies, batch effects exist between bulk RNA-seq and scRNA-seq data. When transferring cellular relationships inferred from bulk RNA-seq to scRNA-seq settings, these inherent batch effects can introduce a graph-level domain shift, thereby reducing the stability of cross-domain predictions.

In this work, we developed DAGFormer (Domain Adaptation Graph-based Transformer), a DL framework that integrates graph transformer architecture with domain adaptation techniques to fuse bulk RNA-seq and single-cell RNA-seq data for predicting drug responses at the single-cell level. DAGFormer is designed to construct biologically meaningful intercellular relationship graphs, while mitigating domain shifts at the graph level between bulk RNA-seq and scRNA-seq data. DAGFormer highlights the following aspects:(i) To the best of our knowledge, DAGFormer is a rare DL model to

explore and access different methods for constructing cellular-cellular interaction network topologies in predicting cancer drug response at the single-cell level through integration with bulk RNA-seq data; (ii) in order to eliminate the inherent batch effects between bulk and single-cell RNA-seq data, DAGFormer employs a graph-based domain adaptation (GDA) mechanism with a domain discriminator and feature extractor in an adversarial training scheme, enabling the model to learn domain-invariant representations; (iii) in order to preserve both shared and domain-specific biological signals, DAGFormer introduces a dual-domain collaborative decoupling-fusion framework, where a shared encoder extracts common features aligned with scRNA-seq characteristics, and private encoders retain domain-specific biological variations from bulk and single-cell data. We conduct extensive experiments on multiple scRNA-seq datasets treated with different drugs. The results demonstrate that DAGFormer outperforms existing models in predicting drug responses at the single-cell level. Overall, we believe that DAGFormer can enhance the accuracy of drug sensitivity predictions and facilitate drug repurposing in cancer treatment.

## 2 Materials and methods

### 2.1 Problem definition

This paper addresses the binary classification task of predicting single-cell cancer drug response. Our primary goal is to leverage rich label information from large-scale cell line data (Bulk RNA-seq) to accurately forecast drug sensitivity at the single-cell level.

To model the input data and the knowledge transfer challenge, we define the data as a graph-structured network $\mathcal{N}$:

$$\mathcal{N} = (G, X, Y)$$

where $G$ is the normalized adjacency matrix of a graph constructed based on sample gene expression similarity, describing the relationships between nodes. $X$ is the node feature matrix containing the standardized gene expression profiles for each sample (node). $Y$ is the node label vector, containing cancer drug response labels (sensitive or resistant), which exists only in the source domain.

Specifically, we define two distinct networks:

1. **Source Domain Network:** $\mathcal{N}_s = (G_s, X_s, Y_s)$
   - $G_s$: The cellular neighbor graph constructed from the similarity of gene expression features of cell lines (Bulk RNA-seq data).
   - $X_s$: The gene expression attribute matrix where each node represents a cell line.
   - $Y_s$: The drug response labels for the cell lines (e.g., sensitive or resistant).
2. **Target Domain Network:** $\mathcal{N}_t = (G_t, X_t)$
   - $G_t$: The cellular neighbor graph constructed from the similarity of gene expression features of single cells (scRNA-seq data).
   - $X_t$: The gene expression attribute matrix where each node represents a single cell.
   - Crucially, the target domain network lacks drug response labels ($Y$).

Our objective is to learn a cross-modality node representation from the richly labeled source network $\mathcal{N}_s$, enabling accurate prediction of drug response labels in the target network $\mathcal{N}_t$. However, the fundamental difference in sequencing technologies and data granularity (Bulk vs. Single-cell) leads to a significant domain shift in both the feature distribution ($X_s$ vs. $X_t$) and the graph topology ($G_s$ vs. $G_t$). Therefore, effectively transferring label knowledge and bridging this graph-level domain gap without relying on target domain labels ($Y_t$) is the central challenge of this work.

## 2.2 Datasets

The source domain data is sourced from the GDSC (Genomics of Drug Sensitivity in Cancer) database. The source domain is primarily used for model training and the learning of drug response knowledge. It is typically a broad and readily accessible dataset, such as the GDSC, which includes data from a large number of cell lines and drug responses. This dataset undergoes preprocessing using RMA normalization, followed by the transformation of IC50 values (half-maximal inhibitory concentration) into binary labels through a logistic optimization binary classification algorithm (LOBIco). These binary labels indicate whether a cell line is classified as 'sensitive' or 'resistant' to a particular drug. Detailed characteristics of the dataset are provided in Table 1, and further information on the dataset can be found in the work by Iorio et al. [2]

The target domain data is derived from multiple scRNA-seq datasets, which are employed for predicting cancer drug responses at the single-cell level. The target domain is typically used for practical application or evaluation. Its data characteristics, such as the use of single-cell RNA sequencing (scRNA-seq), may differ from the source domain. The target domain provides higher-resolution information at the single-cell level, but drug response data is often more scarce. As a result, the knowledge learned from the source domain is transferred to help make predictions in the target domain. The CCLE dataset, provided by the Broad Institute, includes 10x Genomics sequencing data for the JHU006 and SCC47 cell lines, which are utilized to assess drug sensitivities to various compounds such as Gefitinib, Afatinib, AR-42, Cetuximab, NVP-TAE684, Sorafenib, and Vorinostat. In addition, data for the PC9 cell line is obtained from the GSE149215 dataset, which consists of untreated sample cells, facilitating the study of drug responses such as those to Etoposide. The A375 and 451Lu cell line data is sourced from the GSE108383 dataset, which employs the SMART-seq platform and focuses specifically on assessing sensitivity to PLX4720. As shown in Table 2, these target domain datasets span a wide array of drugs and cell lines, offering comprehensive support for analyzing drug sensitivity and resistance at the single-cell level.

The single-cell RNA-seq data within the target domain is utilized to evaluate two critical experimental scenarios, which reflect the two primary mechanisms of tumor drug resistance: Acquired Resistance and Inherent Resistance. The first is the Post-treatment (Acquired Resistance) scenario. This approach focuses on evaluating the state of cells after drug exposure, defining sensitivity and resistance by comparing drug-untreated parental cells with drug-tolerant cells that survived the drug treatment. The results for this type of analysis are presented in Fig 4, covering drug responses for agents such as Etoposide and PLX4720. The second is the Pre-treatment (Inherent Resistance) scenario. This approach assesses the state of cells prior to drug exposure, primarily focusing on the analysis and prediction of pre-existing resistant subpopulations within the untreated population. The research results for this scenario are shown in Fig 5, encompassing responses to various drugs including Gefitinib and Cetuximab. Through the classification into these two distinct

**Table 1**. Source domain dataset.

| Drug | Cell line | Dataset | #Res | #Sens | #Gene | Drug treatment |
|---|---|---|---|---|---|---|
| Gefitinib | Pan-Can | GDSC | 714 | 115 | 10610 | before |
| Afatinib | Pan-Can | GDSC | 682 | 150 | 10684 | before |
| AR-42 | Pan-Can | GDSC | 811 | 80 | 10610 | before |
| Cetuximab | Pan-Can | GDSC | 739 | 122 | 10684 | before |
| Etoposide | Pan-Can | GDSC | 811 | 53 | 9738 | after |
| NVP-TAE684 | Pan-Can | GDSC | 358 | 37 | 10684 | before |
| PLX4720 | Pan-Can | GDSC | 746 | 629 | 11937 | after |
| Sorafenib | Pan-Can | GDSC | 362 | 60 | 10684 | before |
| Vorinostat | Pan-Can | GDSC | 774 | 60 | 10610 | before |

**Note:** #Res represents the number of drug-resistant cell lines; #Sens denotes the number of drug-sensitive cell lines; #Gene refers to the number of genes shared between the source and target domains. "Drug treatment" indicates whether the data was obtained before or after the administration of the drug.

**Table 2**. Target domain dataset.

| Drug | Cell line | Dataset | Platform | #Res | #Sens | #Gene | Drug treatment |
|---|---|---|---|---|---|---|---|
| Gefitinib | JUH006 | CCLE | 10x Genomics | 33 | 33 | 10610 | before |
| Afatinib | SCC47 | CCLE | 10x Genomics | 60 | 60 | 10684 | before |
| AR-42 | JUH006 | CCLE | 10x Genomics | 33 | 33 | 10610 | before |
| Cetuximab | SCC47 | CCLE | 10x Genomics | 60 | 60 | 10684 | before |
| Etoposide | PC9 | GSE149215 | 10x Genomics | 764 | 629 | 9738 | after |
| NVP-TAE684 | SCC47 | CCLE | 10x Genomics | 60 | 60 | 10684 | before |
| PLX4720 | 451Lu | GSE108383 | SMART-seq | 63 | 63 | 11937 | after |
| PLX4720 | A375 | GSE108383 | SMART-seq | 48 | 62 | 11937 | after |
| Sorafenib | SCC47 | CCLE | 10x Genomics | 60 | 60 | 10684 | before |
| Vorinostat | JUH006 | CCLE | 10x Genomics | 33 | 33 | 10610 | before |

**Note:** #Res represents the number of drug-resistant single cells; #Sens indicates the number of drug-sensitive single cells; and #Gene refers to the number of genes shared between the source and target domains.

experimental scenarios, we are able to gain a deeper, single-cell level understanding of the mechanisms underlying different types of drug resistance.

## 2.3 Data preprocessing

The sensitivity and resistance distribution of cell lines under different drug treatments often exhibit discrepancies, leading to the class imbalance problem in drug response labels within the training set. This imbalance can negatively affect the predictive performance of the model. Therefore, when training cancer drug response prediction models at the bulk level, we employed a sequential data balancing pipeline combining SMOTE oversampling and Random UnderSampler to achieve a near 1:1 ratio between sensitive and resistant cell lines. Specifically, this sequential process ensures a robust equilibrium by controlling intermediate ratios: First, we used SMOTE oversampling to generate synthetic minority class samples (sensitive cell lines), bringing their count to 50% of the current majority class count. Second, we used Random UnderSampler to reduce the number of majority class samples (resistant cell lines), ensuring the new majority class count became 50% of the new minority class count. This two-step process effectively mitigates the class imbalance problem in the training set. For the gene expression data processing, we standardized the data using the StandardScaler from scikit-learn, transforming it into a Z-score with a mean of 0 and a standard deviation of 1. We then selected cell lines that had both RMA normalized RNA-seq data and LOBICO binary drug response IC50 values, which served as samples for subsequent analysis. After standardization, the resulting matrix had each row representing a cell line, and each column representing a standardized expression value for a gene. These gene expression data formed the source domain feature matrix used for model training.

The target domain scRNA-seq data underwent quality control for cells and genes based on the recommendations from the literature [34,36]. Specifically, cells with an extremely low number of genes and genes detected in fewer than three cells were filtered out. Additionally, cells with more than 20% mitochondrial gene expression were removed. Furthermore, the total molecular count matrix of the cells was normalized, followed by logarithmic transformation and z-score standardization. After standardization, the resulting matrix had each row representing a cell and each column representing a standardized expression value for a gene. These gene expression data formed the target domain feature matrix used for model training. In addition, to ensure compatibility between the source and target domains, this method retained only genes that were present in both the source and target domain datasets. This process ensures that the model simultaneously utilizes gene features from both the source and target domains, enabling knowledge transfer between the two different datasets.

 

## 2.4 Workflow overview of DAGFormer

The workflow of DAGFormer is illustrated in Fig 1. DAGFormer is a framework combining graph transformers and Graph-based Domain Adaptation (GDA) mechanisms to predict cancer drug responses at the single-cell level. It effectively transfers label information from the source domain (bulk RNA-seq) to the target domain (scRNA-seq) through a dual-domain collaborative decoupling-fusion framework. DAGFormer operates through six key steps: (i) Constructing a cellular neighbor graph for both bulk RNA-seq and scRNA-seq data; (ii) Capturing domain-specific features $H_s^{pr}$ and $H_t^{pr}$ using private encoders; (iii) Establishing biological connections and capturing shared features $H_s^{sh}$ and $H_t^{sh}$ through shared encoders; (iv) Reconstructing the original graphs $\tilde{G}_s$ and $\tilde{G}_t$ with decoders; (v) Mitigating domain shifts between the two domains using adversarial training between the domain discriminator and shared encoder through GDA; and (vi) Using a drug response predictor to forecast cancer drug response at the single-cell level. By leveraging these computational strategies, DAGFormer effectively addresses batch effects in cross-domain analyses, enhancing the accuracy of cancer drug response predictions.

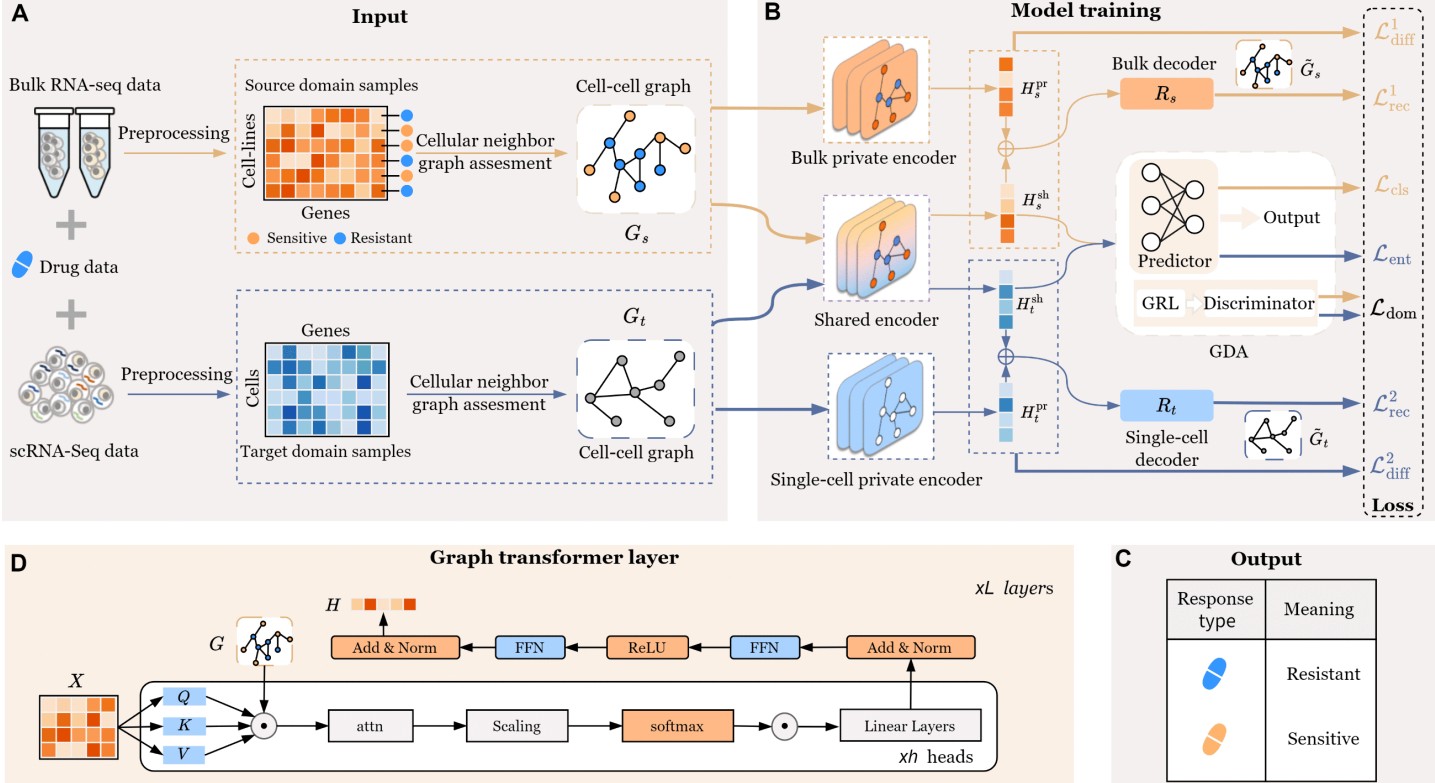

**Fig 1. Overview of DAGFormer.** A: Assessment strategy for constructing cellular neighbor graphs from bulk RNA-seq and scRNA-seq data. B: Dual-domain collaborative decoupling-fusion framework with graph-based domain adaptation (GDA), consisting of five steps: (i) Use private encoders for bulk and single-cell data to extract domain-specific features $H_s^{pr}$ and $H_t^{pr}$; (ii) Use shared encoders to extract common features $H_s^{sh}$ and $H_t^{sh}$ between domains; (iii) Reconstruct graphs $\tilde{G}_s$ and $\tilde{G}_t$ using decoders; (iv) Apply GDA with adversarial training to reduce batch effects and domain shift; (v) Predict single-cell drug response using the integrated features. C: Single-cell drug response prediction via integration with bulk RNA-seq data. D: Architecture of private and shared encoders using graph transformer modules to combine local interactions and global representations.

## 2.5 Exploring cellular neighbor graphs construction

In cancer drug response prediction, intercellular relationships are of significant biological relevance, particularly in terms of cell-cell interactions and communication, cellular metabolic characteristics, and gene expression patterns [38]. However, there is a limited body of research in this field that systematically constructs and explores intercellular relationships using graph structures.

To address this gap, we explore three distinct methods for constructing cellular neighbor graphs, offering a novel perspective for drug response prediction: Spearman Rank Correlation Coefficient (SRCC), Pearson Correlation Coefficient (PCC), and K-Nearest Neighbors (KNN). These methods quantify cell similarities based on gene expression patterns from different biostatistical viewpoints, providing a variety of strategies for creating biologically meaningful cell graphs.

SRCC is a non-parametric method used to assess the monotonic relationship between two continuous variables. In this study, for bulk RNA-seq data, these variables represent the gene expression levels of two cell lines, while for single-cell RNA-seq data, they represent the gene expression levels of two individual cells. SRCC captures the similarity between cells by measuring the rank-order correlation of their gene expression profiles. Specifically, we calculate the SRCC $\rho_{ij}^{SRCC}$ for cells $i$ and $j$ using the following formula:

$$\rho_{ij}^{\text{SRCC}} = \frac{\frac{1}{n}\sum_{k=1}^{n}\left(R(x_k) - \overline{R(x)}\right)\cdot\left(R(y_k) - \overline{R(y)}\right)}{\sqrt{\left(\frac{1}{n}\sum_{k=1}^{n}\left(R(x_k) - \overline{R(x)}\right)^2\right)}}$$
$$\cdot \frac{1}{\sqrt{\left(\frac{1}{n}\sum_{k=1}^{n}\left(R(y_k) - \overline{R(y)}\right)^2\right)}} \tag{1}$$

Where $R(x_k)$ and $R(y_k)$ denote the ranks of gene $k$ in cell $i$ and cell $j$, respectively, and $\overline{R(x)}$ and $\overline{R(y)}$ represent the mean ranks in the corresponding cells. The total number of genes is denoted as $n$.

Using the computed SRCC matrix $\rho^{\text{SRCC}}$, we select the top 20% of cell pairs with the highest correlations as the edges of the graph. The threshold for selection is determined by the percentile, with the 20th percentile used here. The minimum value $\theta$, corresponding to the top 20% of correlations in the coefficient matrix, is then computed. This value serves as the criterion for connecting edges in the graph, thereby constructing the cellular neighbor graph $\mathbf{G}^{\text{SRCC}}$:

$$G_{ij}^{\text{SRCC}} = \begin{cases} 1, & \text{if } \rho_{ij}^{\text{SRCC}} > \theta \\ 0, & \text{otherwise} \end{cases} \tag{2}$$

PCC provides an alternative method to quantify the linear correlation of gene expression between cells. This method assumes that the data follows a normal distribution and is suitable for analyzing the direct linear relationship of gene expression data between cells. The calculation formula for PCC is:

$$\rho_{ij}^{\text{PCC}} = \frac{\sum_{k=1}^{n}(x_{ik} - \overline{x_i})(x_{jk} - \overline{x_j})}{\sqrt{\sum_{k=1}^{n}(x_{ik} - \overline{x_i})^2}\cdot\sqrt{\sum_{k=1}^{n}(x_{jk} - \overline{x_j})^2}} \tag{3}$$

Where $x_{ik}$ and $x_{jk}$ represent the expression levels of the $k$-th gene in cells $i$ and $j$, respectively; $\overline{x_i}$ and $\overline{x_j}$ are the average gene expression values in cells $i$ and $j$, respectively; and $n$ is the total number of genes.

Similar to the SRCC method, we compute the minimum value $\theta$ corresponding to the top 20% of the most correlated cell pairs in the Pearson correlation coefficient matrix $\rho^{PCC}$. These top-correlated pairs are selected as the edges of the graph, thereby constructing the cellular neighbor graph $\mathbf{G}^{PCC}$.

K-Nearest Neighbors (KNN) is a distance-based algorithm that captures similarity between cells by comparing their gene expression patterns. It identifies the $K$ most similar neighbors for each cell based on their proximity in feature space. This enables the construction of biologically meaningful graphs that reflect cell functions and states. In our approach, we calculate cosine similarity between gene expression vectors to determine neighborhood relationships. Specifically, we adopt the `knn_graph` function from the *Deep Graph Library (DGL)* with `dist='cosine'`, where cosine distance is defined as $1 - $ cosine similarity. Cosine similarity is computed as:

$$\text{sim}(x_i, x_j) = \frac{x_i \cdot x_j}{\|x_i\| \, \|x_j\|} \tag{4}$$

Where $x_i$ and $x_j$ denote the gene expression vectors of cells $i$ and $j$, respectively; $x_i \cdot x_j$ is their dot product; and $\|x_i\|$, $\|x_j\|$ are their Euclidean norms. This metric emphasizes directional similarity and ignores magnitude differences, making it particularly effective for high-dimensional gene expression data.

Rather than using similarity values as edge weights, we utilize them to determine connectivity: each cell connects to its $K$ most similar neighbors, forming the cellular neighbor graph $\mathbf{G}^{KNN}$, defined as:

$$G_{ij}^{KNN} = \begin{cases} 1, & \text{if } j \text{ is among the } K \text{ nearest neighbors of } i \\ 0, & \text{otherwise} \end{cases} \tag{5}$$

To ensure the undirected nature of the graph and reduce its sparsity, we apply symmetric normalization to the adjacency matrix. We also perform empirical evaluations to determine the optimal number of neighbors $K$ for the KNN graph construction. Specifically, we compare performance across several values of $K$, including 5, 10, 15, and 30. Based on the experimental results, we select $K = 15$, which yields the best predictive performance in downstream drug response prediction tasks. For implementation, the cellular neighbor graphs ($G_s$ and $G_t$) for both the bulk and single-cell populations are constructed statically (i.e., once prior to any training iteration) and are subsequently utilized in a full-batch manner throughout the entire optimization process to preserve the maximal integrity of cellular relationships.

## 2.6 Dual-domain collaborative decoupling-fusion strategy

Current approaches employing Graph Domain Adaptation (GDA) for drug response prediction predominantly concentrate on learning domain-invariant representations between source (bulk RNA-seq) and target (scRNA-seq) domains. However, these methods frequently fail to account for the distinctive domain-specific features inherent to each data modality. This oversight potentially constrains the model's capacity to discern domain-specific variations, consequently compromising transfer learning performance. Drawing inspiration from the adversarial separation network developed by by Zhang et al. [35], we implement a dual-domain collaborative decoupling-fusion framework. This innovative architecture distinctly separates the modeling of domain-specific features from both bulk and single-cell data, enabling simultaneous capture of both domain-specific and shared biological characteristics.

The integration of private and shared encoders within our framework effectively accomplishes two pivotal objectives: (i) ensuring cross-domain consistency during feature fusion, and (ii) retaining the distinctive biological characteristics inherent to each domain. This dual-domain encoding mechanism achieves an optimal equilibrium between feature alignment and domain-specific preservation, thus addressing the common issue of information loss due to excessive feature alignment encountered in traditional domain adaptation methodologies. Specifically, the strategy encompasses three essential

components: bulk and single-cell private encoders, bulk and single-cell shared encoder, and graph transformer layers for effective local-global feature extraction.

**2.6.1 Private encoders for bulk and single-cell data.** As shown in Fig 1, the source encoder corresponds to the bulk private encoder mentioned above, while the target encoder represents the single-cell private encoder. Specifically, the input gene expression matrices are $X_s \in \mathbb{R}^{N_b \times d}$ and $X_t \in \mathbb{R}^{N_s \times d}$, where $N_b$ and $N_s$ represent the number of cell lines and single cells, respectively, and $d$ denotes the number of genes. Each row of the matrix corresponds to the gene expression profile of a cell, while each column represents the expression value of a gene. After applying z-score normalization, the scale differences between genes are removed. Meanwhile, we construct the graph $G \in \mathbb{R}^{N \times N}$ using the previously developed cellular neighbor graph construction method. For all subsequent analyses and training reported, we utilized a full-batch approach to ensure the entire cellular neighbor graph and feature matrices fit into memory and were processed simultaneously in every epoch.

The graph transformer-based encoder extracts features from the normalized data, where the source and target domains are independently processed by the source encoder (bulk private encoder) and the target encoder (single-cell private encode), respectively, as shown below:

$$H_s^{\text{pr}} = E_s(X_s, G_s), \quad H_t^{\text{pr}} = E_t(X_t, G_t) \tag{6}$$

Where $E_s(\cdot)$ denotes the source domain's private bulk encoder module, $E_t(\cdot)$ represents the target domain's single-cell private encoder module, and $G_s$ and $G_t$ are the cellular neighbor graphs for the source and target domains, respectively. The outputs $H_s^{\text{pr}}$ and $H_t^{\text{pr}}$ represent the private features extracted from the source and target domains.

**2.6.2 Graph transformer layers for effective local-global feature extraction.** To effectively extract informative representations from bulk RNA-seq and scRNA-seq data, we design a multi-layer stacked graph transformer. This transformer takes as input the gene expression profiles of cells and the corresponding cellular neighbor graph, simultaneously modeling local interactions and progressively capturing global feature structures. Taking the Source Encoder as an example, the input gene expression matrix $X_s$ is first projected linearly as follows:

$$H^{(0)} = \text{ReLU}\left(X_s W_{\text{proj}}\right), \tag{7}$$

where the projection matrix $W_{\text{proj}} \in \mathbb{R}^{d \times d_h}$ maps the input dimension $d$ to the hidden dimension $d_h$.

Subsequently, the initial features $H^{(0)}$ are processed by multiple graph transformer layers. A single-layer transformer primarily facilitates local information exchange, but as layers stack, node information is progressively propagated to more distant nodes, achieving global feature integration. At each layer $l$, a sparse multi-head attention mechanism captures node interactions by computing queries ($Q$), keys ($K$), and values ($V$):

$$Q = H^{(l)} W_Q, \quad K = H^{(l)} W_K, \quad V = H^{(l)} W_V, \tag{8}$$

where $W_Q, W_K, W_V \in \mathbb{R}^{d_h \times d_k}$, and $d_k$ is the dimension per attention head.

The *query vector* $Q_i$ is derived from the *target node* $i$, while the *key* $K_j$ and *value* $V_j$ vectors are derived from its *neighboring nodes* $j \in \mathcal{N}(i)$. The attention weights are computed by comparing the target node's query $Q_i$ with each neighbor's key $K_j$, and then used to weight the corresponding value $V_j$, which contains the neighbor's feature information to be aggregated into node $i$'s updated representation.

The attention weights are computed using the dot product between the query and key and are sparsified by the cellular neighbor graph $\hat{G}$. This constraint restricts interactions to biologically relevant neighboring cells, reducing interference

from distant, unrelated nodes:

$$\text{attn}_{ij}^k = \hat{G}_{ij} \cdot \left( Q_i^k \cdot K_j^k \right), \tag{9}$$

where $\text{attn}_{ij}^k$ denotes the similarity score between cell line $i$ and its neighboring cell line $j$ in the $k$-th attention head. The adjacency constraint $\hat{G}_{ij}$ ensures that attention is computed only for directly connected neighbors.

Node representations are updated by aggregating neighbor features using normalized attention weights, employing multi-head concatenation for diversity:

$$H_i^{(l+1)} = \Big\|_{k=1}^{H} \sum_{j \in \mathcal{N}(i)} \text{softmax}_j(\text{attn}_{ij}^k) V_j^k, \tag{10}$$

where $\mathcal{N}(i)$ denotes node $i$'s neighbors, and $\|_{k=1}^{H}$ denotes concatenation across $H$ heads.

To enhance training stability and feature expressiveness, residual connections, batch normalization, and feed-forward networks are utilized:

$$\hat{H}^{(l+1)} = \text{BatchNorm}(H^{(l+1)} + H^{(l)}), \tag{11}$$

$$H_i^{(l+1)} = \text{ReLU}(\hat{H}_i^{(l+1)} W_1 + b_1) W_2 + b_2. \tag{12}$$

Through these stacked layers, the graph transformer efficiently integrates local community interactions with the global structure, ensuring robust feature learning across both bulk and single-cell data modalities.

**2.6.3 Shared encoder for feature alignment.** In addition to capturing the domain-specific features of Bulk RNA-seq and scRNA-seq data, the model must also extract shared feature representations between the two domains. Learning shared features is essential, as it establishes potential biological connections between Bulk and Single-cell data, enabling better cross-domain transfer and knowledge integration, thereby improving the performance of cancer drug response prediction. To address this, we designed a shared encoder for Bulk and Single-cell data, which captures the shared feature representations between the domains by integrating common patterns from both.

Similar to the private encoders, the shared encoder is built upon a graph transformer module. However, its primary role is to learn domain-invariant features by jointly processing gene expression profiles and cellular neighbor graphs from both the source and target domains. Through a domain alignment mechanism, the shared encoder facilitates the extraction of shared representations while preserving the structural information of each domain. The processing of the shared encoder can be expressed as:

$$H_s^{\text{sh}} = E_{\text{sh}}(X_s, G_s), \quad H_t^{\text{sh}} = E_{\text{sh}}(X_t, G_t) \tag{13}$$

Where $E_{\text{sh}}(\cdot)$ denotes the shared encoder module, and $G_s$ and $G_t$ are the cellular neighbor graphs for the source and target domains, respectively. The outputs $H_s^{\text{sh}}$ and $H_t^{\text{sh}}$ represent the shared features extracted from the source and target domains.

Although domain adaptation facilitates the alignment of bulk and single-cell RNA-seq representations, the modality gap between these data types, such as differences in expression granularity and in the graph structures derived from each modality, can introduce domain-specific biases into the shared feature representations and impair the model's generalization. For example, the cellular relationships constructed from bulk data often reflect averaged or smoothed expression patterns, whereas those from single-cell data capture fine-grained heterogeneity. This discrepancy in graph topology further complicates cross-domain learning when using graph-based encoders such as graph transformers.

To mitigate the impact of modality-specific biases, we introduce a difference loss based on soft subspace orthogonality. This constraint encourages the model to learn generalizable transcriptional patterns across domains, while preserving

bulk-specific expression trends and single-cell heterogeneity in separate subspaces, thereby enhancing both predictive performance and biological interpretability. The difference loss $\mathcal{L}_{\text{diff}}$ penalizes the projection overlap between shared and private features from both domains:

$$\mathcal{L}_{\text{diff}} = \left\| H_s^{\text{sh}^\top} H_s^{\text{pr}} \right\|_F^2 + \left\| H_t^{\text{sh}^\top} H_t^{\text{pr}} \right\|_F^2 \tag{14}$$

$$\min_{E_s^{\text{pr}}, E_t^{\text{pr}}, E_{\text{sh}}} \mathcal{L}_{\text{diff}} = \min_{E_s^{\text{pr}}, E_t^{\text{pr}}, E_{\text{sh}}} \left( \mathcal{L}_{\text{diff}}^1 + \mathcal{L}_{\text{diff}}^2 \right) \tag{15}$$

Where $\| \cdot \|_F^2$ denotes the Frobenius norm, which measures the similarity between shared and private features. $H_s^{\text{sh}}$ and $H_t^{\text{sh}}$ represent the shared features for the source and target domains, respectively, while $H_s^{\text{pr}}$ and $H_t^{\text{pr}}$ denote the private features for each domain. By minimizing $L_{\text{diff}}$, the model ensures that the shared features $H_s^{\text{sh}}$ and $H_t^{\text{sh}}$ contain only the common feature patterns from both Bulk and Single-cell data, while domain-specific information is preserved independently in the private features $H_s^{\text{pr}}$ and $H_t^{\text{pr}}$.

## 2.7 Decoding and graph reconstruction

The decoder in DAGFormer is designed to reconstruct biologically meaningful cellular neighbor graphs by integrating both domain-specific and shared representations. Biologically, this design enables the model to preserve cell-type-enriched transcriptional features in single-cell RNA-seq data and tissue-level averaged profiles in bulk RNA-seq, while capturing common transcriptional programs shared across modalities. Technically, the decoder adopts an Inner Product Decoder architecture. For each domain, the decoder receives a concatenation of domain-specific features (output from the private encoder) and domain-invariant features (output from the shared encoder). As shown in Fig 1, the decoder responsible for the source domain reconstructs the graph derived from bulk RNA-seq data, while the target-domain decoder reconstructs the graph derived from single-cell RNA-seq data. For example, the decoder for the source domain $D_s$ reconstructs the source graph $\tilde{G}_s$ from the concatenated features $H_s^{\text{pr}}$ and $H_s^{\text{sh}}$, while the target domain decoder $D_t$ reconstructs the target graph $\tilde{G}_t$ from $H_t^{\text{pr}}$ and $H_t^{\text{sh}}$:

$$\tilde{G}_s = R_s \left( [H_s^{\text{pr}}, H_s^{\text{sh}}] \right), \quad \tilde{G}_t = R_t \left( [H_t^{\text{pr}}, H_t^{\text{sh}}] \right) \tag{16}$$

Here, $[\cdot, \cdot]$ denotes the concatenation operation along the feature dimension. $\tilde{G}_s$ and $\tilde{G}_t$ are the reconstructed cellular neighbor graphs for the source and target domains, respectively, and $R_s$ and $R_t$ represent the corresponding inner product decoding functions.

To ensure that the reconstructed cellular neighbor graphs accurately preserve the topological structures of the original graphs, we introduce a reconstruction loss composed of binary cross-entropy terms, separately computed for the source and target domains:

$$\mathcal{L}_{\text{rec}}^1 = -\mathbb{E}_{(i,j) \sim G_s} \left[ g_{ij}^s \log \tilde{g}_{ij}^s + (1 - g_{ij}^s) \log(1 - \tilde{g}_{ij}^s) \right] \tag{17}$$

$$\mathcal{L}_{\text{rec}}^2 = -\mathbb{E}_{(i,j) \sim G_t} \left[ g_{ij}^t \log \tilde{g}_{ij}^t + (1 - g_{ij}^t) \log(1 - \tilde{g}_{ij}^t) \right] \tag{18}$$

Here, $g_{ij}^s$ and $g_{ij}^t$ denote the entries in the ground-truth adjacency matrices for the source and target domains, respectively, while $\tilde{g}_{ij}^s$ and $\tilde{g}_{ij}^t$ are the corresponding predicted values from the reconstructed graphs $\tilde{G}_s$ and $\tilde{G}_t$. The overall reconstruction loss is defined as:

$$\min_{E_s^{\text{pr}}, E_t^{\text{pr}}, R_s, R_t} \mathcal{L}_{\text{rec}} = \min_{E_s^{\text{pr}}, R_s} \mathcal{L}_{\text{rec}}^1 + \min_{E_t^{\text{pr}}, R_t} \mathcal{L}_{\text{rec}}^2 \tag{19}$$

## 2.8 Graph-based domain adaptation (GDA) via adversarial training

To reduce domain shift between bulk RNA-seq (source) and single-cell RNA-seq (target) data, we introduce a graph-based domain adaptation (GDA) module based on adversarial learning. The goal is to align shared feature representations $H_s^{\text{sh}}$ and $H_t^{\text{sh}}$ across domains, promoting domain invariance and improving cross-modality generalization. Specifically, DAGFormer employs a domain discriminator combined with a Gradient Reversal Layer (GRL). The domain discriminator $D(\cdot)$, structured as a two-layer fully connected neural network, distinguishes whether the input shared features originate from bulk RNA-seq data or scRNA-seq data. Conversely, the shared encoder $E_{\text{sh}}(\cdot)$ aims to produce features indistinguishable by the discriminator, thereby generating domain-invariant feature representations.

The GRL passes features unchanged during forward propagation but reverses the sign of gradients and dynamically scales their magnitude during backpropagation. This encourages the shared encoder to iteratively refine the features towards domain invariance, making it increasingly difficult for the discriminator to distinguish the domain origin of the features.

The adversarial training process utilizes a domain adversarial loss $\mathcal{L}_{\text{dom}}$, formally defined as:

$$\mathcal{L}_{\text{dom}} = \mathbb{E}_{x^s \sim \Phi^s}[\log D(H_s^{\text{sh}})] + \mathbb{E}_{x^t \sim \Phi^t}[\log(1 - D(H_t^{\text{sh}}))], \tag{20}$$

where $\Phi^s$ and $\Phi^t$ denote the distributions of the bulk RNA-seq and scRNA-seq data, respectively. Through this adversarial optimization, the domain discriminator aims to maximize $\mathcal{L}_{\text{dom}}$, enhancing its ability to differentiate domains, while the shared encoder attempts to minimize $\mathcal{L}_{\text{dom}}$, ensuring feature invariance.

## 2.9 Drug response predictor

The Drug Response Predictor utilizes the shared encoder module $E_{\text{sh}}$ to extract drug response information from bulk RNA-seq data and transfer it to the scRNA-seq data, thereby enabling drug response prediction at the single-cell level. The drug response predictor $P(\cdot)$ is a simple fully connected layer. It incorporates classification loss in the labeled source domain network and entropy loss in the unlabeled target domain network, allowing it to learn feature representations that effectively distinguish between drug-sensitive and drug-resistant labels.

In the source domain, the drug response predictor applies a linear transformation to the shared features $H_s^{\text{sh}}$ to generate the drug sensitivity probability $\psi(x_i^s)$. Specifically, $\psi(x_i^s)$ represents the probability that cell line $i$ is predicted to be drug-sensitive (label = 1), whereas $1 - \psi(x_i^s)$ denotes the probability that it is predicted to be drug-resistant (label = 0). The objective of the drug response predictor is to minimize the discrepancy between the predicted values and the true labels, which is achieved using the cross-entropy loss function, defined as:

$$\mathcal{L}_{\text{cls}} = -\frac{1}{N_s} \sum_{i=1}^{N_s} \left[ y_i^s \log \psi(x_i^s) + (1 - y_i^s) \log\left(1 - \psi(x_i^s)\right) \right] \tag{21}$$

where $y_i^s \in \{0, 1\}$ represents the ground-truth drug response label in the source domain, $N_s$ denotes the number of cell lines in the source domain, and $L_{\text{cls}}$ is the cross-entropy loss function applied to the drug response classification task.

In the target domain, due to the absence of ground-truth drug response labels, DAGFormer introduces an entropy loss term to enhance the predictive capability of the drug response predictor on unlabeled data. This is achieved by minimizing the conditional entropy of the predicted probabilities in the target domain:

$$\mathcal{L}_{\text{ent}} = -\frac{1}{N_t} \sum_{i=1}^{N_t} \left[ \psi(x_i^t) \log \psi(x_i^t) + (1 - \psi(x_i^t)) \log\left(1 - \psi(x_i^t)\right) \right] \tag{22}$$

where $\psi(x_i^t)$ represents the probability that single cell $i$ in the target domain is predicted to be drug-sensitive (label = 1), whereas $1 - \psi(x_i^t)$ denotes the probability that it is predicted to be drug-resistant (label = 0). $N_t$ represents the number of single cells in the target domain.

## 2.10 Overall objective function and optimization

To facilitate knowledge transfer from the source domain to the target domain, the total loss function integrates three primary objectives, as described in the previous sections. First, it extracts both private and shared feature representations from bulk RNA-seq and scRNA-seq data. Second, it learns discriminative features for drug response classification, enabling the classification of bulk RNA-seq data. Third, it employs a graph-adaptive mechanism to extract domain-invariant feature representations.The overall objective function is defined as:

$$\mathcal{L} = \mathcal{L}_{\text{cls}} + \lambda_d \mathcal{L}_{\text{dom}} + \lambda_e \mathcal{L}_{\text{ent}} + \lambda_f \mathcal{L}_{\text{diff}} + \lambda_r \mathcal{L}_{\text{rec}}, \tag{23}$$

where the coefficients $\lambda_d$, $\lambda_e$, $\lambda_f$, and $\lambda_r$ control the relative contributions of each loss term, ensuring that the model appropriately balances the different objectives during optimization.

# 3 Results

## 3.1 Baselines

To assess the effectiveness of our proposed model, we compare it against the following baseline approaches:

- **SCAD** [34] uses bulk RNA-seq data as the source domain and single-cell RNA-seq data as the target domain. It learns shared features between the source and target domains through three modules: the feature extractor, the drug response predictor, and the domain discriminator. The model leverages transfer learning and adversarial domain adaptation techniques, optimizing the feature extractor by combining binary cross-entropy loss with adversarial loss functions. This enables accurate prediction of drug responses at the single-cell level.
- **scDEAL** [33] employs a domain-adaptive neural network (DaNN) with denoising autoencoders to harmonize bulk and single-cell RNA-seq data, aligning their feature spaces through maximum mean discrepancy (MMD) loss and cell-cluster regularization. This deep transfer learning framework transfers drug-response patterns from bulk data to predict single-cell-level drug sensitivity.
- **scAdaDrug** [4]: A deep transfer learning model that predicts drug sensitivity using Adversarial Multi-source Domain Adaptation. It aligns feature vectors derived from multiple source domains, focusing on reducing the distributional distance between feature embeddings.
- **scGSDR** [5]: A recent pharmacological profiling model that enhances feature quality by harnessing gene semantics and pathway knowledge. It improves prediction by utilizing a highly informed, high-quality feature vector representation derived from biological prior knowledge.
- **Random Forest** is an ensemble learning algorithm based on multiple decision trees. It trains several trees on subsets of the data and outputs the majority vote for classification. We use it as a non-deep baseline to predict drug response labels from gene expression data.
- **Logistic Regression** is a classic linear model for binary classification. It learns a linear decision boundary by modeling the probability of class membership using a sigmoid function, and serves as a strong interpretable baseline.
- **Support Vector Machine (SVM)** is a margin-based classifier that finds an optimal hyperplane to separate classes in the feature space. Both linear and non-linear kernels are tested for drug response classification.
- **Decision Tree** builds a hierarchical tree structure by recursively splitting features based on impurity measures. It is easy to interpret and provides a simple baseline for performance comparison.

- **Naive Bayes** is a probabilistic classifier based on Bayes' theorem, assuming feature independence.
- **AdaBoost** is a boosting-based ensemble method that combines weak learners into a strong classifier by focusing on misclassified examples.
- **XGBoost (Extreme Gradient Boosting)**: A highly optimized gradient boosting framework that is renowned for its speed and performance in classification tasks.
- **LightGBM (Light Gradient Boosting Machine)**: Another highly efficient and scalable gradient boosting framework that uses gradient-based one-side sampling and exclusive feature bundling to handle large datasets.

### 3.2 Evaluation metrics

Our model and all the baselines methods are tested on the CCLE, GSE149215, and GSE108383 datasets respectively. we measure the predictive performance by using accuracy (ACC), area under ROC curve (AUC) and area under precision/recall curve (AUPR) .

- **ACC** is used to evaluate the overall correctness of the model's predictions, defined as the ratio of correctly predicted samples to the total number of samples.
- **AUC** reflects the model's classification performance across various thresholds. It is computed based on the ROC curve, which plots the true positive rate on the vertical axis and the false positive rate on the horizontal axis. A higher AUC value, closer to 1, indicates better model performance. We used the 'roc_auc_score' function from the sklearn library to implement AUC evaluation.
- **AUPR** calculates the relationship between precision and recall, and measures the model's performance in handling imbalanced data by the area under the precision-recall curve. AUPR represents the average precision of the model across all possible recall values. We used the 'average_precision_score' function from the sklearn library to implement AUPR evaluation.

### 3.3 Cellular neighbor graph construction strategies comparison

In this subsection, we conducted 9 experiments to evaluate drug responses on different compounds: Gefitinib, Afatinib, AR-42, Cetuximab, Etoposide, NVP-TAE684, PLX4720, Sorafenib, and Vorinostat. Notably, PLX4720 was evaluated on the A375 cell line. For each drug, the model was trained using bulk RNA-seq data from the GDSC database and tested on scRNA-seq datasets from CCLE, GSE149215, and GSE108383, using various graph construction strategies.

Fig 2D summarizes the performance of three graph construction strategies—PCC, SRCC, and KNN—across these drugs, evaluated using ACC, AUC, and AUPR. Overall, the KNN-based approach achieves the most robust and consistent performance, with average ACC, AUC, and AUPR scores of 0.937, 0.969, and 0.976, respectively. These values are higher than those of PCC (0.881, 0.943, 0.938) and SRCC (0.737, 0.822, 0.829). The violin plots demonstrate that KNN yields more concentrated performance distributions, indicating greater stability and generalizability across different drug contexts.

From a biological perspective, the superior performance of KNN likely stems from its ability to preserve local transcriptional neighborhoods among cells, which often reflect functional cell states or drug-responsive subpopulations that are critical for capturing differential sensitivity at the single-cell level. Furthermore, the selection of the parameter $K = 15$ for the KNN graph was determined through a dedicated empirical ablation study. We systematically evaluated the model's performance (ACC, AUC, and AUPR) across a range of tested $K$ values (5, 10, 15, and 30) across all ten drug datasets. The results, detailed in Fig 2, confirmed that $K = 15$ achieves the highest overall average performance. This optimal value successfully balances graph sparsity (low $K$ leading to insufficient connectivity) with graph density (high $K$ introducing extraneous noise), thereby maximizing the quality of the learned graph embeddings. In contrast, global correlation-based strategies like PCC and SRCC focus on overall expression trends across the dataset, making them susceptible to being

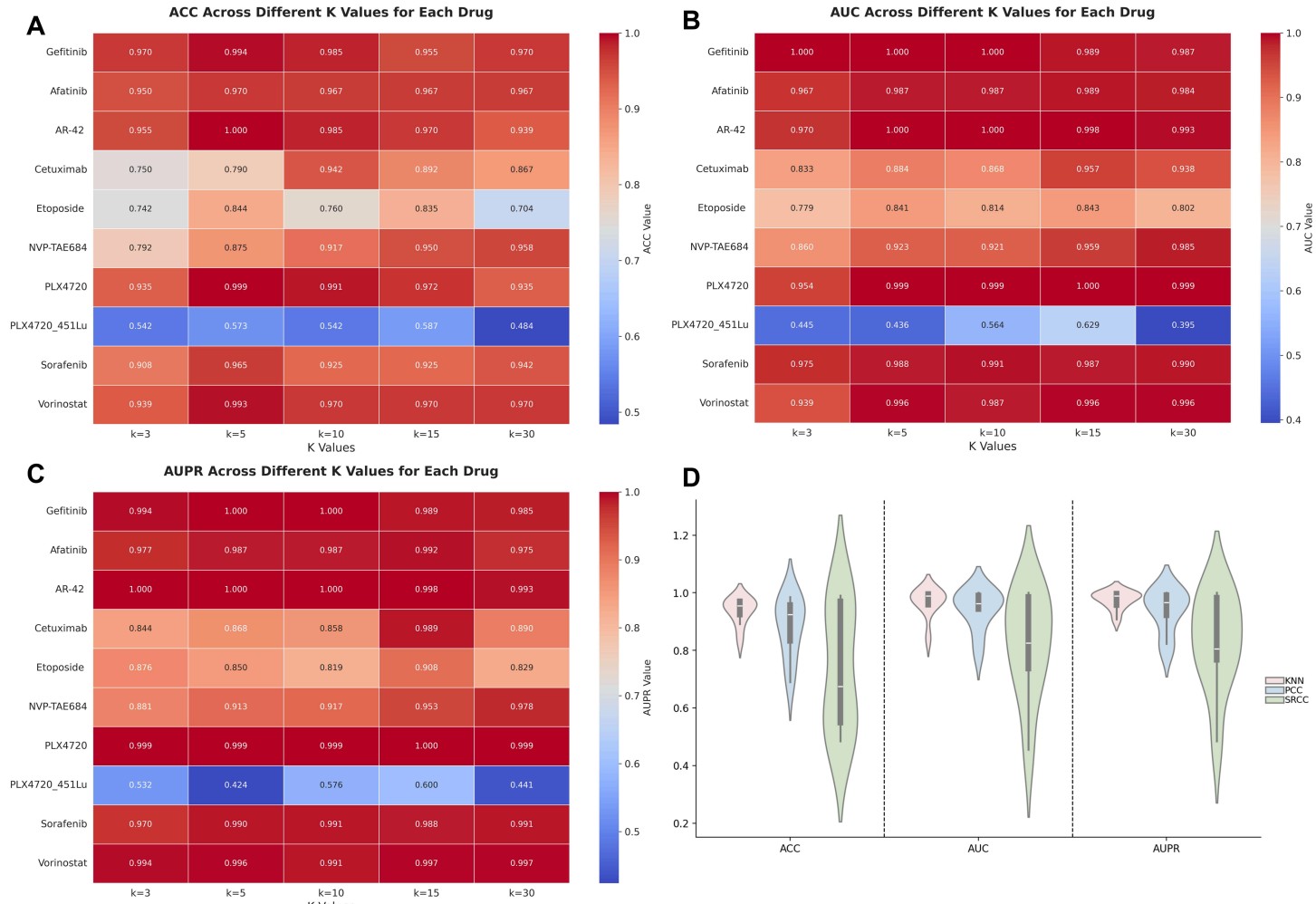

**Fig 2**. Comprehensive evaluation of DAGFormer performance based on K-value selection and graph construction strategies.

dominated by highly expressed genes and potentially masking the coordinated activity of lowly expressed but functionally important genes. As a result, PCC and SRCC may overlook fine-grained transcriptional patterns, particularly in datasets with high cellular heterogeneity.

However, we also observed interesting exceptions. In the case of Etoposide, the PCC-based graph outperformed KNN by a margin of approximately 0.1 in AUC and AUPR, suggesting that the transcriptional responses to this drug may follow a global, linear co-expression pattern. Biologically, Etoposide is a topoisomerase II inhibitor that induces DNA damage, and its downstream effects may involve coordinated expression of DNA repair genes or stress-response modules. Such broad, synchronous expression patterns are more effectively captured by PCC, which measures global linear correlations.

The cellular neighbor graph construction is the most computationally demanding step, dominated by the $O(N^2 \cdot g)$ time complexity required for distance calculation and KNN identification, where $N$ is the number of samples (cells/cell lines) and $g$ is the number of shared genes. Since this high-complexity process is pre-computed once prior to model training, the total time cost remains manageable (typically taking minutes for our moderate $N$). The storage burden is also critical due to our full-batch training strategy. While the gene expression features ($X$) require $O(N \cdot g)$ storage, the graph adjacency matrix is stored in a highly efficient sparse format, resulting in a minimal storage complexity of $O(N \cdot K)$ (where $K$ is the number of neighbors).

## 3.4 Benchmarking DAGFormer against classical and deep models

In this section, we conducted 10 experiments to evaluate drug responses on different drugs: Gefitinib, Afatinib, AR-42, Cetuximab, Etoposide, NVP-TAE684, PLX4720, Sorafenib, and Vorinostat. It is noteworthy that PLX4720 was tested in two separate experiments on the A375 and 451Lu cell lines. For each drug, we trained the model using the bulk RNA-seq data as the source domain and used the scRNA-seq data to evaluate the model's performance.

To better contextualize our method, we briefly summarize the core mechanisms of the two main deep learning baselines, scDEAL and SCAD, both of which aim to transfer drug response knowledge from bulk RNA-seq to single-cell RNA-seq data. scDEAL employs two modality-specific Denoising Autoencoders to independently extract low-dimensional features from bulk and single-cell expression matrices. A domain-adaptive neural network aligns these features using a MMD loss, enforcing distribution similarity in the embedding space. Notably, scDEAL uses two separate encoders without any shared components, meaning feature alignment occurs only at the embedding level, with no enforced consistency during feature extraction. SCAD introduces a shared encoder architecture alongside adversarial domain adaptation. It employs a domain discriminator and a gradient reversal layer (GRL) to promote domain-invariant shared features. This adversarial training strategy offers greater robustness compared to the MMD-based alignment in scDEAL. However, SCAD still treats gene expression profiles as flat feature vectors, ignoring cellular interactions and topological structures—factors that are biologically critical for modeling drug response at the single-cell level.

As shown in Fig 3, we compared DAGFormer against a suite of baseline methods, including classical machine learning models (e.g., Random Forest, Logistic Regression, SVM), ensemble-based learners (e.g., AdaBoost), and state-of-the-art DL frameworks (scDEAL, SCAD variants). DAGFormer consistently outperforms all competing models across all drugs, achieving the best performance in terms of ACC, AUC, and AUPR. To ensure a rigorous and fair comparison, and given the complexity of replicating fine-grained preprocessing steps, the results for the SCAD [34] were directly adopted from the original publication. This decision guarantees that DAGFormer is benchmarked against SCAD operating at its optimal, validated configuration. Crucially, our own dataset processing pipeline, including Z-score standardization and

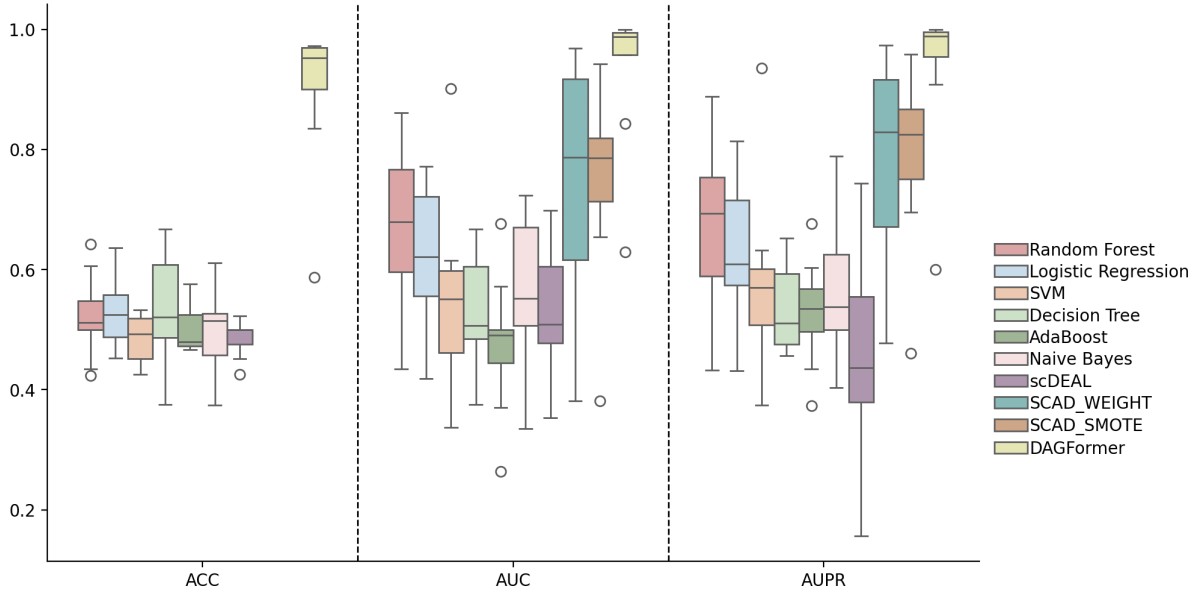

**Fig 3**. **Performance comparison of methods on various drugs.**

Quality Control standards for both Bulk and scRNA-seq data, was intentionally designed to mirror the rigorous preprocessing methodology used in the original SCAD study, ensuring the input feature spaces are highly congruent. However, since the SCAD model only reports AUC and AUPR as evaluation metrics, the ACC values are missing in the corresponding results. Notably, DAGFormer exhibits a tightly concentrated distribution near the upper bound across AUC and AUPR metrics, indicating both high predictive accuracy and low inter-dataset variance. In contrast, traditional models show broader distributions with lower medians, reflecting limited capacity to capture the heterogeneous response patterns among single cells.

Among DL baselines, SCAD achieves relatively strong performance due to its use of adversarial domain adaptation, which explicitly promotes the learning of domain-invariant features while preserving domain-specific information. In contrast, scDEAL demonstrates inferior performance in our benchmark (as shown in Fig 3), largely because its alignment strategy—based solely on minimizing the MMD—lacks a dedicated mechanism to disentangle shared and domain-specific representations. This simplistic alignment in the embedding space may fall short under complex batch effects or nonlinear shifts, limiting its robustness in real-world settings. As shown in Figs 4 and 5, SCAD models ranked second only to DAGFormer, achieving average AUC and AUPR values of approximately 0.80 and 0.83, respectively. DAGFormer further extends this line of improvement by performing domain adaptation at the graph level. Unlike SCAD, which aligns flat feature vectors, DAGFormer aligns node representations within cellular interaction graphs. This allows it to capture both domain-invariant expression profiles and topological similarities between bulk and single-cell modalities. On average, DAGFormer achieves 0.902 ACC, 0.935 AUC, and 0.938 AUPR across ten independent runs—representing relative gains of 13% in AUC and 10% in AUPR over SCAD. These results underscore the importance of modeling cell–cell relationships and demonstrate the effectiveness of graph-based domain adaptation in enhancing single-cell drug response prediction.

**3.4.1 Drug response prediction in drug-tolerant cells after treatment.** Patients can develop drug tolerance relatively quickly due to the emergence of drug-resistant cells after treatment. Consequently, cells that have not been exposed to any drugs, referred to as parental cells, are classified as sensitive cells [39]. In contrast, cells that survive drug exposure are categorized as resistant cells. As shown in Fig 4, we applied transfer learning techniques to two GSE datasets, each containing scRNA-seq data and drug response information from three distinct cell lines. One dataset corresponds to Etoposide, while the other involves PLX4720, a drug used in melanoma treatment by inhibiting the BRAF protein, thus preventing cancer cell growth. The PLX4720 dataset includes two cell lines: 451Lu and A375. For Etoposide, DAGFormer outperformed the baseline models, achieving the highest values in all metrics—ACC (0.835), AUC (0.843), and AUPR (0.908)—demonstrating its effectiveness in predicting drug sensitivity at the single-cell level.

In the case of PLX4720, DAGFormer exhibited markedly different predictive performance between the 451Lu and A375 cell lines. For 451Lu, regardless of the model used, both AUC and AUPR were close to 0.5, indicating almost random predictions and suggesting that the model could not effectively distinguish between sensitive and resistant cells in this cell line. In contrast, for A375, the model achieved perfect results, with AUC = 1.000 and AUPR = 1.000. As shown in Fig 6, the sensitive and resistant samples in A375 are inherently well-separated in the transcriptomic expression space, enabling the model to easily learn a clear decision boundary. The relatively small sample size for A375 (only 110 cells) may have further contributed to this perfect score.

The stark difference in performance between A375 (AUC= 1.000) and 451Lu (AUC≈ 0.60) is directly driven by the inherent transcriptional separability of the sensitive and resistant cell populations in each cell line. This critical factor dictates the model's ability to learn a reliable decision boundary. Previous work by Ho et al. [37] provides biological insight into this performance gap. Their clustering analysis of scRNA-seq profiles revealed that, within each cell line, untreated (*parental*) cells group closely with their BRAFi-resistant counterparts: untreated 451Lu cells cluster with BRAFi-resistant 451Lu cells, and untreated A375 cells cluster with BRAFi-resistant A375 cells. This means that for each cell line, the gene expression profiles of sensitive and resistant cells are quite similar, while the differences between cell lines are even greater than the changes caused by drug treatment. In other words, cell-type-specific transcriptomic identity can dominate

| Method | Etoposide ACC | AUC | AUPR | PLX4720_A375 ACC | AUC | AUPR | PLX4720_451Lu ACC | AUC | AUPR | Mean Score ACC | AUC | AUPR |
|---|---|---|---|---|---|---|---|---|---|---|---|---|
| AdaBoost | 0.548 | 0.501 | 0.562 | 0.472 | 0.430 | 0.570 | 0.523 | 0.497 | 0.499 | 0.514 | 0.476 | 0.543 |
| XGBOOST | 0.452 | 0.482 | 0.540 | 0.426 | 0.837 | 0.845 | 0.516 | 0.416 | 0.441 | 0.465 | 0.578 | 0.609 |
| LightGBM | 0.452 | 0.593 | 0.623 | 0.426 | 0.848 | 0.864 | 0.516 | 0.392 | 0.399 | 0.465 | 0.611 | 0.629 |
| Random Forest | 0.424 | 0.441 | 0.505 | 0.435 | 0.861 | 0.888 | 0.523 | 0.435 | 0.433 | 0.461 | 0.579 | 0.609 |
| Logistic Regression | 0.453 | 0.546 | 0.580 | 0.556 | 0.769 | 0.814 | 0.497 | 0.418 | 0.432 | 0.502 | 0.578 | 0.609 |
| Decision Tree | 0.524 | 0.496 | 0.547 | 0.639 | 0.632 | 0.652 | 0.490 | 0.485 | 0.470 | 0.551 | 0.538 | 0.556 |
| Naive Bayes | 0.444 | 0.550 | 0.549 | 0.611 | 0.724 | 0.789 | 0.374 | 0.335 | 0.403 | 0.476 | 0.536 | 0.580 |
| SCAD_SMOTE | - | 0.694 | 0.736 | - | 0.825 | 0.875 | - | 0.381 | 0.461 | - | 0.633 | 0.691 |
| SCAD_WEIGHT | - | 0.669 | 0.694 | - | 0.694 | 0.769 | - | 0.381 | 0.478 | - | 0.581 | 0.647 |
| scGSDR | 0.595 | 0.652 | 0.633 | 0.857 | 0.964 | 0.957 | 0.581 | 0.542 | 0.617 | 0.677 | 0.719 | 0.736 |
| scDEAL | 0.452 | 0.476 | 0.407 | 0.426 | 0.617 | 0.427 | 0.523 | 0.484 | 0.546 | 0.467 | 0.526 | 0.460 |
| scAdaDrug | 0.607 | 0.629 | 0.669 | 0.727 | 0.812 | 0.894 | 0.562 | 0.559 | 0.529 | 0.632 | 0.667 | 0.697 |
| DAGFormer | 0.835 | 0.843 | 0.908 | 0.972 | 1.000 | 1.000 | 0.587 | 0.629 | 0.600 | 0.798 | 0.824 | 0.836 |

**Fig 4**. **Performance comparison of different methods across drugs (post-treatment).** The best results are highlighted with the darkest green circular markers. The top four results are indicated by green circles with progressively deeper shades, where darker colors represent better performance. SCAD variants differ in sampling strategies: SCAD_WEIGHT employs weighted sampling and SCAD_SMOTE uses SMOTE sampling. '–' indicates unavailable data. All values represent the mean performance across three independent trials. Cellular neighbor graphs in DAGFormer were constructed using the KNN method (k=15).

over the drug-induced transcriptional shift. For 451Lu, this intrinsic similarity between sensitive and resistant states likely makes it difficult for the model to learn a clear separation, resulting in near-random predictions. In contrast, in A375, the baseline transcriptomic profile is inherently closer to the sensitive state, making sensitive and resistant populations more distinct and easier for the model to classify.

**3.4.2 Drug response prediction for pre-existing drug-resistant cells before treatment.** Although single-cell drug response prediction has gained increasing attention, few studies have systematically assessed the feasibility of using DL to predict drug sensitivity at the single-cell level before treatment [40]. This is particularly important, as mounting evidence suggests that pre-existing drug-resistant subpopulations within tumors can respond differently to therapy, driving treatment failure, tumor relapse, and poor clinical outcomes [41,42]. Therefore, accurately characterizing intratumoral heterogeneity before treatment is critical, and scRNA-seq—owing to its single-cell resolution—offers a more effective approach than bulk RNA-seq in this context.

As shown in Fig 5, we selected seven clinically relevant anticancer drugs and evaluated model performance using cell lines from the GDSC database as the source domain (Table 1) and JUH006 and SCC47 cell lines as the target domain (Table 2). The results demonstrate that DAGFormer consistently achieves state-of-the-art performance across all seven drugs, outperforming both traditional machine learning models and the most recent deep learning baselines. The non-deep

| Method | Gefitinib | | | Afatinib | | | AR-42 | | | Cetuximab | | | NVP-TAE684 | | | Sorafenib | | | Vorinostat | | |
|---|---|---|---|---|---|---|---|---|---|---|---|---|---|---|---|---|---|---|---|---|---|
| | ACC | AUC | AUPR | ACC | AUC | AUPR | ACC | AUC | AUPR | ACC | AUC | AUPR | ACC | AUC | AUPR | ACC | AUC | AUPR | ACC | AUC | AUPR |
| AdaBoost | 0.485 | 0.37 | 0.435 | 0.467 | 0.492 | 0.547 | 0.470 | 0.264 | 0.373 | 0.525 | 0.572 | 0.603 | 0.475 | 0.489 | 0.496 | 0.475 | 0.489 | 0.522 | 0.576 | 0.676 | 0.676 |
| XGBOOST | 0.500 | 0.761 | 0.774 | 0.500 | 0.729 | 0.731 | 0.500 | 0.372 | 0.475 | 0.500 | 0.590 | 0.619 | 0.500 | 0.564 | 0.534 | 0.500 | 0.526 | 0.554 | 0.500 | 0.713 | 0.696 |
| LightGBM | 0.500 | 0.736 | 0.723 | 0.500 | 0.602 | 0.597 | 0.500 | 0.386 | 0.444 | 0.500 | 0.612 | 0.609 | 0.500 | 0.546 | 0.539 | 0.500 | 0.782 | 0.786 | 0.500 | 0.782 | 0.786 |
| SVM | 0.530 | 0.466 | 0.526 | 0.533 | 0.512 | 0.581 | 0.485 | 0.600 | 0.602 | 0.467 | 0.460 | 0.501 | 0.500 | 0.458 | 0.502 | 0.525 | 0.594 | 0.600 | 0.500 | 0.615 | 0.632 |
| Naive Bayes | 0.515 | 0.553 | 0.527 | 0.517 | 0.535 | 0.515 | 0.530 | 0.680 | 0.668 | 0.500 | 0.498 | 0.495 | 0.408 | 0.408 | 0.452 | 0.608 | 0.664 | 0.629 | 0.515 | 0.673 | 0.613 |
| Random Forest | 0.606 | 0.694 | 0.742 | 0.642 | 0.770 | 0.774 | 0.500 | 0.762 | 0.718 | 0.541 | 0.664 | 0.668 | 0.550 | 0.617 | 0.613 | 0.500 | 0.589 | 0.581 | 0.500 | 0.768 | 0.757 |
| Logistic Regression | 0.636 | 0.772 | 0.728 | 0.558 | 0.585 | 0.577 | 0.485 | 0.652 | 0.638 | 0.475 | 0.430 | 0.465 | 0.558 | 0.680 | 0.677 | 0.550 | 0.735 | 0.764 | 0.500 | 0.591 | 0.573 |
| Decision Tree | 0.667 | 0.667 | 0.608 | 0.525 | 0.525 | 0.513 | 0.485 | 0.485 | 0.493 | 0.517 | 0.517 | 0.509 | 0.375 | 0.375 | 0.457 | 0.433 | 0.433 | 0.469 | 0.636 | 0.636 | 0.624 |
| scDEAL | 0.500 | 0.683 | 0.708 | 0.467 | 0.353 | 0.157 | 0.500 | 0.531 | 0.370 | 0.500 | 0.468 | 0.447 | 0.500 | 0.698 | 0.744 | 0.500 | 0.568 | 0.558 | 0.500 | 0.486 | 0.283 |
| scAdadrug | 0.800 | 0.840 | 0.878 | 0.583 | 0.725 | 0.720 | 0.500 | 0.550 | 0.541 | 0.534 | 0.798 | 0.784 | 0.437 | 0.734 | 0.821 | 0.388 | 0.339 | 0.420 | 0.550 | 0.590 | 0.638 |
| SCAD_SMOTE | - | 0.773 | 0.824 | - | 0.925 | 0.938 | - | 0.801 | 0.826 | - | 0.798 | 0.843 | - | 0.654 | 0.695 | - | 0.771 | 0.794 | - | 0.942 | 0.958 |
| SCAD_WEIGHT | - | 0.967 | 0.973 | - | 0.880 | 0.889 | - | 0.968 | 0.970 | - | 0.923 | 0.921 | - | 0.598 | 0.664 | - | 0.582 | 0.627 | - | 0.902 | 0.904 |
| scGSDR | 0.769 | 0.952 | 0.948 | 0.833 | 0.923 | 0.920 | 0.786 | 0.896 | 0.938 | 0.833 | 0.871 | 0.833 | 0.792 | 0.821 | 0.868 | 0.833 | 0.832 | 0.890 | 0.769 | 0.929 | 0.959 |
| DAGFormer | 0.955 | 0.989 | 0.989 | 0.967 | 0.989 | 0.992 | 0.970 | 0.998 | 0.998 | 0.892 | 0.957 | 0.958 | 0.950 | 0.959 | 0.953 | 0.925 | 0.987 | 0.988 | 0.970 | 0.996 | 0.997 |

**Fig 5**. **Performance comparison of different methods across drugs (pre-treatment).** The best results are highlighted with the darkest green circular markers. The top four results are indicated by green circles with progressively deeper shades, where darker colors represent better performance. '–' indicates unavailable results due to method-specific constraints. DAGFormer results were obtained using KNN-based graphs with $k = 15$.

learning models, including XGBoost and LightGBM, show poor predictive capability, confirming their inability to capture the high-dimensional complexity of pre-treatment transcriptional profiles.

Among the deep learning baselines, the performance gap between DAGFormer and its strongest competitor, scGSDR, is decisive. For example, for Sorafenib, DAGFormer achieves AUC = 0.987 and AUPR = 0.988, compared to scGSDR's AUC = 0.771 and AUPR = 0.794. Similarly, for NVP-TAE684, DAGFormer improves the AUC by 26.1% and the AUPR by 20.9% over the next-best model (scAdaDrug). Crucially, even against the strong classical domain adaptation competitor SCAD, DAGFormer shows consistent superiority. For Gefitinib, where the SCAD_WEIGHT variant achieves an impressive secondary performance of AUC = 0.967 and AUPR = 0.973, DAGFormer surpasses this result, reaching AUC = 0.989 and AUPR = 0.989. This consistently superior performance is directly attributable to the core innovations of DAGFormer, which integrates the modeling of cell–cell interactions to capture complex topological structures with a Dual-Domain Collaborative Decoupling–Fusion strategy. The latter employs private encoders to preserve unique domain-specific biological signals, while the GDA module aligns the shared, invariant features. Together, these mechanisms form a comprehensive structural and feature-level framework that makes DAGFormer the most robust model for predicting drug resistance.

## 3.5 Evaluating the effectiveness of shared and private encoders in DAGFormer via latent space visualization

To better understand the representational capacity of DAGFormer, we visualized the feature embeddings generated by the raw input, private encoder, and shared encoder using UMAP across five drugs: Cetuximab, Gefitinib, PLX4720,

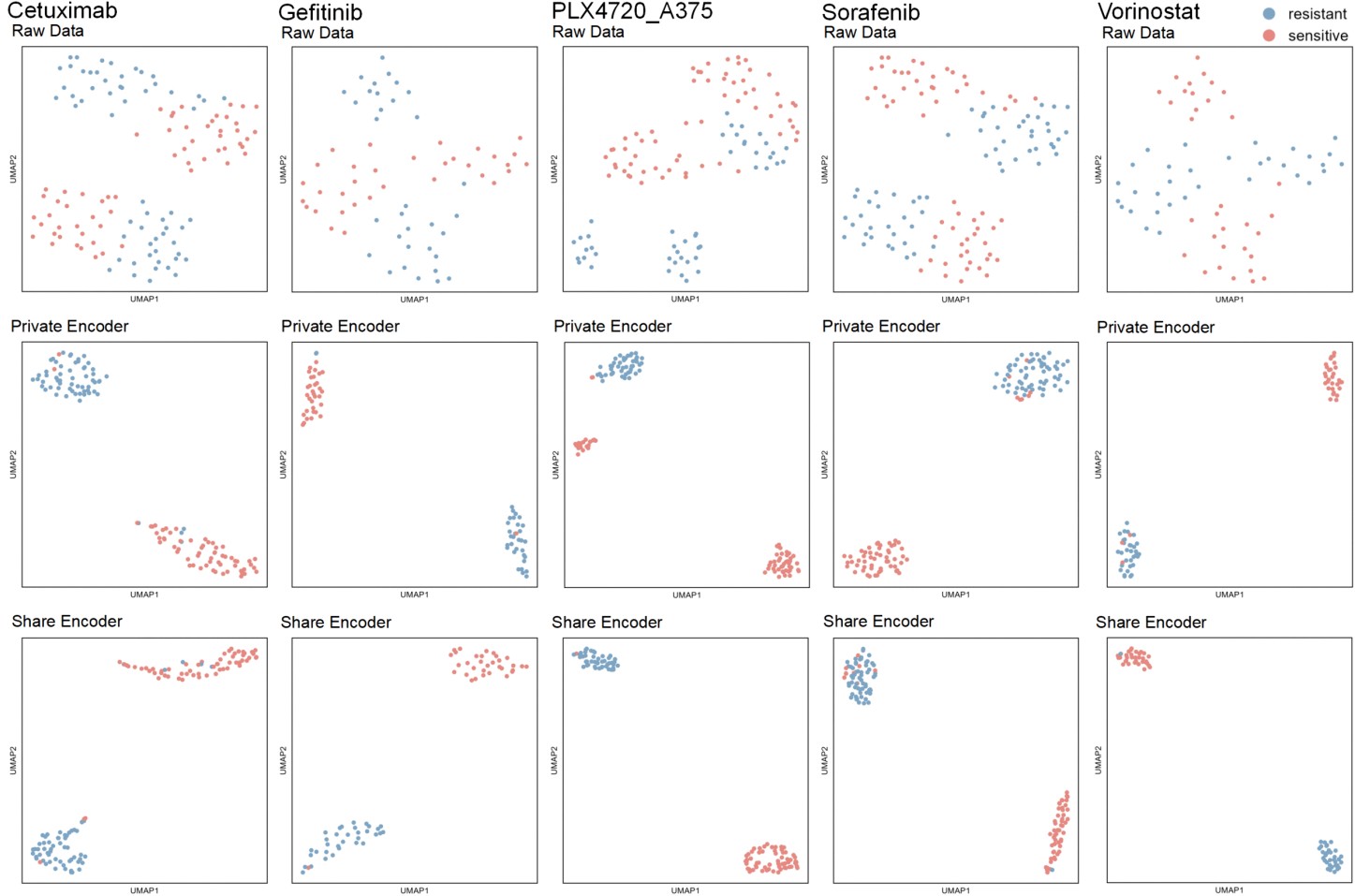

**Fig 6. UMAP visualization of single-cell drug-response representations.** Columns: Cetuximab, Gefitinib, PLX4720, Sorafenib, Vorinostat. Rows: top = raw scRNA-seq profiles; middle = DAGFormer private-encoder embeddings; bottom = shared-encoder embeddings. Each point is a cell, colored by response label (red = sensitive, blue = resistant).

Sorafenib, and Vorinostat (Fig 6). In the raw data (top row), the distribution of sensitive and resistant cells exhibits noticeable overlap, indicating that the original gene expression space does not provide a clear separation between drug response states.

Compared to the raw expression space (top row), where sensitive and resistant cells show partial overlap, both the private (middle row) and shared (bottom row) encoders yield more clearly separated clusters. This visual separation indicates that DAGFormer successfully transforms input data into a representation space where drug response categories are more distinguishable. The consistent separation across drugs and encoder types suggests the model effectively captures information relevant to drug sensitivity in both domain-specific and domain-invariant forms.

### 3.6 Exploring the impact of training set sampling size on DAGFormer performance

To explore the impact of different sampling percentages (20%, 40%, 60%, 80%, and 100%) on the performance of the DAGFormer model in drug response prediction tasks, we selected datasets from Figs 4 and 5 where DAGFormer's accuracy (ACC) was below 0.9. The drugs involved in this study include Cetuximab, Etoposide, and PLX4720 (451Lu), as

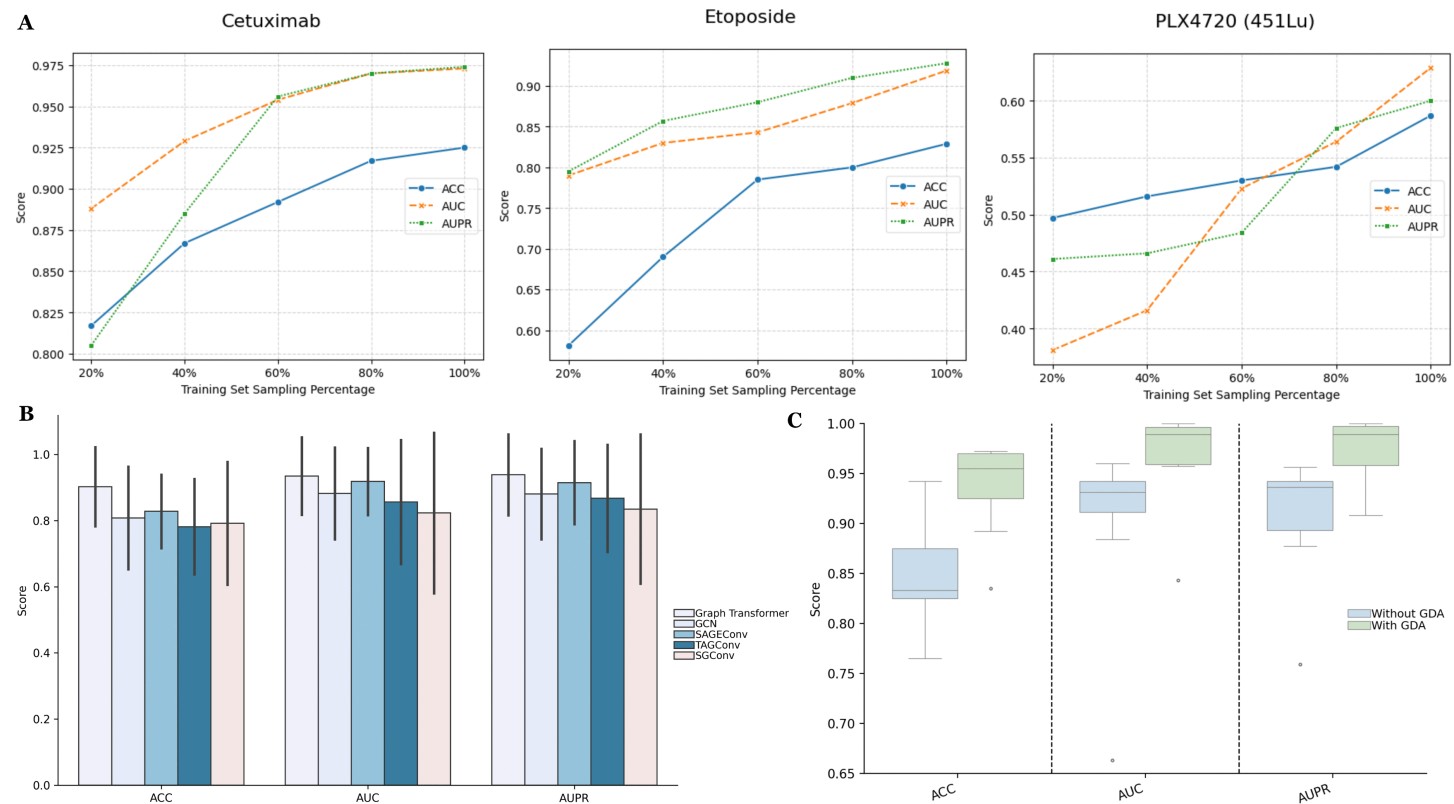

**Fig 7**. **Comprehensive evaluation of DAGFormer.** (A) Performance under varying training set sizes; (B) Comparison with alternative graph encoders; (C) Ablation analysis of the Graph-based Domain Adaptation (GDA) mechanism.

shown in Fig 7A. As observed from the three line plots, with increasing training set size, the DAGFormer model demonstrates significant improvement across all evaluation metrics (ACC, AUC, AUPR) for each drug dataset.

For Etoposide, as the sampling percentage increases from 20% to 100%, ACC improves from 0.581 to 0.829, AUC from 0.790 to 0.919, and AUPR from 0.795 to 0.928. Similarly, for Cetuximab, ACC increases from 0.817 to 0.925, AUC from 0.888 to 0.973, and AUPR from 0.805 to 0.974. However, for PLX4720_451Lu, the improvements are more gradual, with ACC rising from 0.497 to 0.587, AUC from 0.381 to 0.629, and AUPR from 0.461 to 0.600. These results highlight the crucial role of training data size in optimizing model performance. Although the degree of improvement varies across different drugs, the overall trend indicates that an increase in the amount of training data significantly enhances the performance of the model in drug response prediction tasks.

### 3.7 Sensitivity to loss function hyperparameters

The stability and reliability of the proposed GDA framework were rigorously established through a systematic sensitivity analysis focused on the core weighting coefficients ($\lambda_d$, $\lambda_r$, and $\lambda_f$) of the total objective function (Eq 23). This study, conducted on the Gefitinib dataset, validates that the optimal hyperparameter regime is not a narrow, precarious peak, but rather a broad, highly stable region (Fig 8). The tested ranges and default settings are summarized in Table 3.

The results provide critical validation for the high stability of the core GDA components: The model exhibits high performance stability across the entire tested range of the Reconstruction Loss weight ($\lambda_r$), with performance metrics (ACC, AUC, AUPR) remaining near-optimal and showing only marginal peaks around $\lambda_r = 1.0$ (Fig 8A). This robust performance

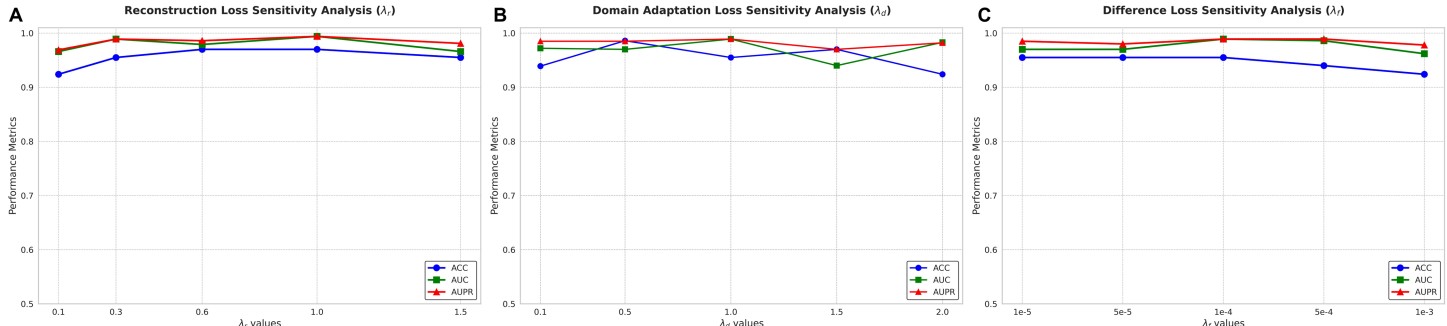

**Fig 8**. Sensitivity analysis of core loss function weighting coefficients ($\lambda_d$, $\lambda_r$, $\lambda_f$).

**Table 3**. Tested ranges for loss function weight coefficients.

| Weight ($\lambda$) | Default Setting | Tested Range |
|---|---|---|
| $\lambda_d$ (DA Loss) | 0.5 | $\{0.1, 0.5, 1.0, 1.5, 2.0\}$ |
| $\lambda_r$ (Reconstruction Loss) | 0.3 | $\{0.1, 0.3, 0.6, 1.0, 1.5\}$ |
| $\lambda_f$ (Difference Loss) | 0.0001 | $\{10^{-5}, 5 \times 10^{-5}, 10^{-4}, 5 \times 10^{-4}, 10^{-3}\}$ |

demonstrates that the overall objective is tolerant to variations in the emphasis placed on structural preservation, confirming that the Graph Transformer encoders effectively integrate feature data and topological structure. Similarly, the constraint that drives Domain Alignment ($\lambda_d$) also shows consistent high performance across a wide range ($\lambda_d \in \{0.1, \dots, 2.0\}$). This resilience to changes in the adversarial pressure confirms the high stability of the shared encoder; the broad, stable plateau observed validates that the default setting operates well within the safe margin, ensuring strong domain invariance without being overly sensitive to the level of alignment pressure. Furthermore, the model is extremely insensitive to changes in the Difference Loss constraint ($\lambda_f$), maintaining highly consistent performance across the tested orders of magnitude ($10^{-5}$ to $10^{-3}$). This stability confirms that the dual-encoder architecture effectively manages feature disentanglement, and the overall system is highly robust against minor tuning of the privacy constraint (Fig 8C). In conclusion, the sensitivity analysis strongly validates the superior robustness and reliability of DAGFormer's architecture, ensuring that its state-of-the-art performance is consistently maintained across practical variations in hyperparameter settings.

In addition, the Entropy Loss weight ($\lambda_e$) was intentionally excluded from the traditional fixed-value grid search because it serves a dynamic stabilization purpose unique to adversarial training. The role of $\lambda_e$ is to aggressively sharpen the decision boundary on the unlabeled target data. Applying this maximum strength too early can destabilize the model by promoting highly confident, but incorrect, initial predictions. Therefore, $\lambda_e$ is controlled by a linear schedule ($\lambda_e(t)$). This dynamic strategy ensures that the entropy constraint gradually intensifies only as the feature alignment and classification knowledge mature, guaranteeing a more robust and stable UDA training progression.

### 3.8 Performance on independent hold-out target data

To rigorously address the concern regarding the generalization capacity of our Unsupervised Domain Adaptation (UDA) framework, particularly when utilizing the entire target dataset for unsupervised training, we conducted a critical ablation experiment to evaluate DAGFormer's performance on a completely independent, unseen subset of single-cell data. This study validates the robustness of our GDA mechanism and its capability for effective knowledge transfer. We selected the Etoposide dataset, which has $N_t = 1,393$ single cells (derived from 764 resistant and 629 sensitive cells, according to Table 2), due to its substantial single-cell count, enabling a reliable data split. The original Etoposide scRNA-seq data ($\mathcal{N}_t$)

was partitioned into a Training Target Set ($\mathcal{N}_t^{\text{train}}$) and a strictlyIndependent Hold-out Set ($\mathcal{N}_t^{\text{holdout}}$), using an 80%/20% split. During the training phase, only the $\mathcal{N}_t^{\text{train}}$ set was used to calculate all unsupervised losses ($\mathcal{L}_{adv}$, $\mathcal{L}_{diff}$, $\mathcal{L}_{rec}$, $\mathcal{L}_{ent}$), ensuring the $\mathcal{N}_t^{\text{holdout}}$ data was entirely excluded from contributing to any model loss.

The evaluation results comparing the original UDA scheme with the new ablation scheme are presented in Table 4. The analysis confirms the successful generalization capability of our GDA mechanism while highlighting the nuanced role of the Entropy Loss ($\mathcal{L}_{ent}$). Crucially, the domain-invariant metrics, AUC and AUPR, showed relatively small declines of only 0.040 ($0.843 \rightarrow 0.803$) and 0.053 ($0.908 \rightarrow 0.855$), respectively, demonstrating that the model maintained a strong absolute predictive performance on the entirely unseen data. This outcome emphatically confirms the core effectiveness of our method: the Shared Encoder, optimized by the GDA mechanism, learns highly generalized domain-invariant features that retain powerful discriminative ability and reliability, even when a significant portion of the target domain samples is strictly withheld from training. The more pronounced drop in ACC of 0.107 ($0.835 \rightarrow 0.728$) is primarily attributed to the reduction in the operational benefit of the $\mathcal{L}_{ent}$ term. In the original scheme, $\mathcal{L}_{ent}$ utilized all target domain data to aggressively sharpen the decision boundary by minimizing the uncertainty (entropy) of predictions, which subsequently boosted the overall ACC score. By excluding 20% of the data from the training set, the model loses the benefit of this boundary sharpening mechanism on the held-out samples. This observed decrease in ACC thus reflects the loss of the high-accuracy gain provided by the density maximization function itself, rather than indicating a fundamental failure of the feature alignment or knowledge transfer mechanisms. In summary, this ablation study validates that DAGFormer's learned feature representation is inherently robust and generalizable across the domain gap, while $\mathcal{L}_{ent}$ provides a beneficial, data-dependent enhancement to prediction confidence.

## 3.9 Ablation study

To evaluate the contributions of the different modules of our model to the prediction performance, we conduct an ablation study by comparing five variants of the model:

- **Graph Convolutional Network (GCN):** A method that aggregates features from neighboring nodes using spectral convolution to extract features from the feature matrix and cell-cell graph.
- **Graph Convolutional Network (SAGEConv):** A method that uses neighbor sampling to aggregate information for inductive learning, extracting features from the feature matrix and cell-cell graph.
- **Targeted Graph Convolution (TAGConv):** A method that applies targeted aggregation techniques, focusing on specific nodes or regions to extract features from the feature matrix and cell-cell graph.
- **Simplified Graph Convolution (SGConv):** A simplified version of graph convolution that reduces computational complexity while extracting features from the feature matrix and cell-cell graph.
- **Graph Transformer (Ours):** A method that uses attention mechanisms to focus on relevant neighbors, extracting features from the feature matrix and cell-cell graph.

These variants' performances and our model are shown in Fig 7B. The results demonstrate that the Graph Transformer outperforms the GCN, SAGEConv, TAGConv, SGConv, and Graph Attention Network across all three evaluation metrics.

**Table 4**. Performance comparison on etoposide dataset: original UDA vs. independent hold-out ablation.

| Metric | Original UDA Scheme | New Ablation Scheme | Performance Decline (Δ) |
|--------|---------------------|---------------------|-------------------------|
|        | (All $\mathcal{N}_t$ used in UDA Training) | (20% $\mathcal{N}_t$ Held Out) | |
| ACC    | 0.835 | 0.728 | 0.107 |
| AUC    | 0.843 | 0.803 | 0.040 |
| AUPR   | 0.908 | 0.855 | 0.053 |

This highlights the superior ability of the Graph Transformer to extract features and effectively predict drug responses in comparison to other variants.

In order to further validate the effectiveness of the proposed GDA mechanism, we conducted an ablation study by comparing the performance of DAGFormer with and without the GDA mechanism across nine different drug datasets, including Gefitinib, Afatinib, AR-42, Cetuximab, Etoposide, NVP-TAE684, PLX4720, Sorafenib, and Vorinostat. Notably, the test for PLX4720 was performed using the A375 cell line.

As shown in Fig 7C, the model with GDA (yellow boxes) consistently outperforms the model without GDA (blue boxes) across all three metrics. The GDA-enhanced model not only achieves higher median scores but also exhibits a smaller interquartile range, indicating improved performance and stability. Specifically, the average scores for the model with GDA are ACC = 0.937, AUC = 0.969, and AUPR = 0.976, compared to ACC = 0.826, AUC = 0.899, and AUPR = 0.907 for the model without GDA. The integration of GDA results in performance gains of 13.4% ($\Delta$ACC=0.111), 7.8% ($\Delta$AUC=0.070), and 7.6% ($\Delta$AUPR=0.069), demonstrating its effectiveness.

While the baseline model performs poorly on PLX4720 drug response prediction (AUC=0.663, AUPR=0.759), the GDA-enhanced model shows minimal performance degradation on Etoposide data (ACC=0.835, AUC=0.843). This confirms that the GDA mechanism effectively mitigates domain shift issues while maintaining robust predictive capabilities across drug responses at the single-cell level. The improvements in both central tendency and dispersion metrics strongly support the efficacy of our GDA approach in enhancing model generalizability.

## 4 Conclusion

In this paper, we propose DAGFormer,a GDA model for cancer drug response prediction. The model constructs an inter-cell similarity graph using three distinct strategies to capture the complex relationships between cells. To enable the adaptive integration of bulk RNA-seq and scRNA-seq data, DAGFormer employs a GDA mechanism, utilizing adversarial training between a domain discriminator and feature map transformation encoder. This approach effectively reduces the distributional differences between the two data types. To balance "feature alignment" with "domain specificity retention" and avoid the loss of feature information due to excessive alignment in traditional domain adaptation methods, DAGFormer introduces a shared encoder and private encoders. This design ensures that, during the feature fusion process, data consistency is preserved while retaining the unique biological information from each data source. Experimental results demonstrate that DAGFormer outperforms existing methods across multiple datasets, including CCLE.

## 5 Discussion

While DAGFormer achieves state-of-the-art results, this study is subject to several limitations that suggest avenues for future research. Firstly, cancer drug responses are influenced by multi-modal data, including transcriptomic, epigenomic, and proteomic data. This study considers only transcriptomic data, and the integration of multi-modal data is highly challenging due to differences in measurement scales, data structures, and noise levels. Secondly, most existing methods treat drug responses as binary classification tasks, converting IC50 values into binary labels. This approach overlooks the continuous and complex nature of drug responses. Future research should shift the task from binary classification to regression to more accurately predict continuous drug response values and capture finer distinctions in cellular responses.

## Author contributions

**Conceptualization:** ZhiHua Du.

**Data curation:** Fen Yan.

**Formal analysis:** Fen Yan.

**Funding acquisition:** Fen Yan, ZhiHua Du, Yu-An Huang.

**Investigation:** Fen Yan.

**Methodology:** Fen Yan, Yu-An Huang.

**Project administration:** ZhiHua Du, Yu-An Huang.

**Resources:** ZhiHua Du.

**Software:** Fen Yan.

**Supervision:** ZhiHua Du, Yu-An Huang.

**Validation:** Fen Yan.

**Visualization:** Fen Yan.

**Writing – original draft:** Fen Yan.

**Writing – review & editing:** ZhiHua Du, Yu-An Huang.

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
