## [Decision Letter · Decision Letter 0]

7 Oct 2025

PCOMPBIOL-D-25-01609

DAGFormer: A graph-based domain adaptation approach for single-cell cancer drug response prediction

PLOS Computational Biology

Dear Dr. Huang,

Thank you for submitting your manuscript to PLOS Computational Biology. After careful consideration, we feel that it has merit but does not fully meet PLOS Computational Biology's publication criteria as it currently stands. Therefore, we invite you to submit a revised version of the manuscript that addresses the points raised during the review process.

Please submit your revised manuscript within 60 days Dec 07 2025 11:59PM. If you will need more time than this to complete your revisions, please reply to this message or contact the journal office at ploscompbiol@plos.org. Please include the following items when submitting your revised manuscript:

We look forward to receiving your revised manuscript.

Kind regards,

Wei Li, Ph.D.

Academic Editor

PLOS Computational Biology

Pedro Mendes

Section Editor

PLOS Computational Biology

**Journal Requirements:**

At this stage, the following Authors/Authors require contributions: ZhiHua Du, Fen Yan, and Yu-An Huang. Please ensure that the full contributions of each author are acknowledged in the "Add/Edit/Remove Authors" section of our submission form.

2) Your manuscript is missing the following section: Discussion.  Please ensure all required sections are present and in the correct order. Make sure section heading levels are clearly indicated in the manuscript text, and limit sub-sections to 3 heading levels. An outline of the required sections can be consulted in our submission guidelines here:

Note: Please label and upload your main figures in numerical order. For Ex: Figure 1,2,3,4,5 so that the reference would be more clear. 

Potential Copyright Issues:

i) Figure 1A. Please confirm whether you drew the images / clip-art within the figure panels by hand. If you did not draw the images, please provide (a) a link to the source of the images or icons and their license / terms of use; or (b) written permission from the copyright holder to publish the images or icons under our CC BY 4.0 license. Alternatively, you may replace the images with open source alternatives. See these open source resources you may use to replace images / clip-art:

3) If any authors received a salary from any of your funders, please state which authors and which funders.

6) We have noticed that you have uploaded references.bib as a separate file. As stated in the PLOS template, your references should be included in your .tex file (not submitted separately as .bib or .bbl). Please also ensure that you are making any formatting changes to both your .tex file and the PDF of your manuscript. If you have any questions, please contact Latex@plos.org. You can find our LaTeX guidelines here: https://journals.plos.org/ploscompbiol/s/latex".

**Reviewers' comments:**

Reviewer's Responses to Questions

**Comments to the Authors:**

**Please note that one review is uploaded as an attachment.**

Reviewer #1: The manuscript by Yan, Du, and Huang presents DAGFormer, an graph-based domain adaptation framework for integrating bulk RNA-seq and scRNA-seq data to predict cancer drug responses at the single-cell level. The prediction of single-cell drug response is crucial for advancing personalized cancer therapy, and the challenges of integrating heterogeneous data sources (bulk vs. single-cell) are well-recognized. The manuscript is well-written and clearly articulates this significance. I recommend acceptance after revisions, as outlined below.

1. In Figure 1, the bulk RNA-seq dataset (Table 1) is used as the source domain and scRNA-seq datasets (Table 2) as the target domain. Please clarify whether the scRNA-seq data are used exclusively for evaluation or also involved during training.

2. The overall loss function includes several weighting coefficients. Please provide details on how these hyperparameters were selected (e.g., grid search, prior work, or manual tuning). In addition, specify the computational environment (e.g., GPU model, hardware configuration) to support reproducibility.

3. Figure 5 visualization: The color scheme of Figure 5 could be improved for consistency and readability. A unified and aesthetically optimized palette would enhance clarity.

4. Minor language issue: On page 3, line 76, “Bulk RNA-seq” is capitalized incorrectly. For consistency, please revise to “bulk RNA-seq” (lowercase, unless at the beginning of a sentence).

Reviewer #2: Uploaded as an attachment

Reviewer #3: The authors propose a Graph-based Domain Adaptation framework that integrates bulk and scRNA-seq data for predicting single-cell drug responses. The DAGFormer builds cell–cell graphs for bulk and single-cell expression (via SRCC, PCC, or KNN), passes them through private (domain-specific) and shared encoders with a graph-transformer backbone to reconstruct graphs and extract the multi-level features, and aligns domains with adversarial (GDA) domain adaptation while predicting single-cell drug response with a trained predictor. Benchmarking DAGFormer on ten independent scRNA-seq datasets demonstrated its superior performance compared to existing methods. However, there are still some limitations.

Major issues:

Q1. There are several hyperparameters for constructing graphs (like SRCC threshold, individual weights in the objective) in the DAGFormer; the current manuscript lacks robustness evaluation, which can reflect the influence of those hyperparameters on the performance of the proposed model.

Q2. The author needs to provide the specific definition of the loss function in the discriminator for the source domain and the target domain.

Q3. There are many models developed in the past two years, but the authors only selected scDEAL, SCAD, and some general machine learning algorithms as baseline methods. It is confusing, and the conclusion would be more comprehensive and reliable if the author compared the proposed model with a more recently developed method.

Q4: The introduction states DAGFormer is "the first DL model" to explore multiple cellular-graph topologies for single-cell drug response with bulk integration; unless a comprehensive survey confirms this, the phrasing should be softened.

Q5: Please provide a baseline protocol table, including versions, seeds, epochs, learning rates, etc.

Q6: The graph construction processing is time-consuming; the author should discuss the proposed models' time complexity and storage burden.

Minor issues:

Q1: The author needs to provide more details about the datasets in the article.

Q2: The GitHub page needs to be improved, and a tutorial is also necessary.

Q3: There are some errors in equation (20), especially the character "where" should not be indented two spaces to the right at the beginning of the paragraph.

Q4: The author needs to improve figure clarity (fonts, resolution, consistent labels).

**Have the authors made all data and (if applicable) computational code underlying the findings in their manuscript fully available?**

Reviewer #1: Yes

Reviewer #2: Yes

Reviewer #3: Yes

PLOS authors have the option to publish the peer review history of their article (what does this mean?). If published, this will include your full peer review and any attached files.

Reviewer #1: No

Reviewer #2: **Yes: **Yangqi Su

Reviewer #3: No

**Figure resubmission:**
---

## [Decision Letter · Decision Letter 1]

10 Dec 2025

Dear dr. Huang,

We are pleased to inform you that your manuscript 'DAGFormer: A graph-based domain adaptation approach for single-cell cancer drug response prediction' has been provisionally accepted for publication in PLOS Computational Biology.

Best regards,

Wei Li, Ph.D.

Academic Editor

PLOS Computational Biology

Pedro Mendes

Section Editor

PLOS Computational Biology

Reviewer's Responses to Questions

**Comments to the Authors:**

Reviewer #1: The authors have revised the manuscript accroding to my comments.

Reviewer #2: Uploaded as an attachment

Reviewer #3: The issues mentioned in the first round review have been addressed.

**Have the authors made all data and (if applicable) computational code underlying the findings in their manuscript fully available?**

Reviewer #1: None

Reviewer #2: Yes

Reviewer #3: Yes

PLOS authors have the option to publish the peer review history of their article (what does this mean?). If published, this will include your full peer review and any attached files.

Reviewer #1: No

Reviewer #2: **Yes: **Yangqi Su

Reviewer #3: No

---

## [Editor Report · Acceptance letter]

PCOMPBIOL-D-25-01609R1

DAGFormer: A graph-based domain adaptation approach for single-cell cancer drug response prediction

Dear Dr Huang,

I am pleased to inform you that your manuscript has been formally accepted for publication in PLOS Computational Biology. Your manuscript is now with our production department and you will be notified of the publication date in due course.

With kind regards,

Anita Estes
